# Learning to Solve Orienteering Problem with Time Windows and Variable Profits

**Songqun Gao**[1,3], **Zanxi Ruan**[4,*], **Patrick Floor**[5], **Marco Roveri**[2,3], **Luigi Palopoli**[2,3],
**Daniele Fontanelli**[1,3]
[1]Department of Industrial Engineering, Università di Trento, Trento, Italy
[2]Department of Information Engineering and Computer Science, Università di Trento, Trento, Italy
[3]Interdepartmental Robotics Labs (IDRA), Trento, Italy
[4]Department of Engineering for Innovation Medicine, Università di Verona, Verona, Italy
[5]Pipple, Eindhoven, Netherlands
`songqun.gao@unitn.it`, `zanxi.ruan@univr.it`

## Abstract

The orienteering problem with time windows and variable profits (OPTWVP) is common in many real-world applications and involves continuous time variables. Current approaches fail to develop an efficient solver for this orienteering problem variant with discrete and continuous variables. In this paper, we propose a learning-based two-stage DEcoupled discrete-Continuous optimization with Service-time-guided Trajectory (DeCoST), which aims to effectively decouple the discrete and continuous decision variables in the OPTWVP problem, while enabling efficient and learnable coordination between them. In the first stage, a parallel decoding structure is employed to predict the path and the initial service time allocation. The second stage optimizes the service times through a linear programming (LP) formulation and provides a long-horizon learning of structure estimation. We rigorously prove the global optimality of the second-stage solution. Experiments on OPTWVP instances demonstrate that DeCoST outperforms both state-of-the-art constructive solvers and the latest meta-heuristic algorithms in terms of solution quality and computational efficiency, achieving up to 6.6x inference speedup on instances with fewer than 500 nodes. Moreover, the proposed framework is compatible with various constructive solvers and consistently enhances the solution quality for OPTWVP.

## 1 Introduction

The Vehicle Routing Problem (VRP) is one of the most fundamental and extensively studied combinatorial optimization problems (COP), aiming to determine which nodes to visit and in what order under given constraints. The orienteering Problem (OP), an important variant of VRP, assumes a fixed total time budget and requires selecting a subset of nodes to visit in order to maximize the total collected reward, which has demonstrated significant potential in practical scenarios such as factory scheduling (Gannouni et al., 2020), logistics transportation (Baty et al., 2024), robotic planning (Ding et al., 2025; Naik et al., 2024), etc. However, in many real-world tasks, the two basic assumptions of the OP often do not hold: fixed rewards and always-accessible nodes. In practice, rewards usually increase with longer service times (variable profit) (Khodadadian et al., 2022; Wan et al., 2024), and many nodes are only accessible within specific time windows (Chen et al., 2024). Thus, the optimizer must decide not only which nodes to visit, but also the service time at each node. These requirements naturally give rise to the Orienteering Problem with Time Windows and Variable Profits (OPTWVP) (Marzal & Sebastia, 2024; Yu et al., 2019), which simultaneously involves discrete route planning and continuous service-time allocation under a fixed time budget to maximize the total reward. Figure 1 (a) illustrates a representative example of OPTWVP applications.

Compared with classical VRPs, OPTWVP represents a more realistic and challenging class of VRP in which discrete routing decisions and continuous service-time allocations are tightly coupled: Unlike

---

*Corresponding authors.

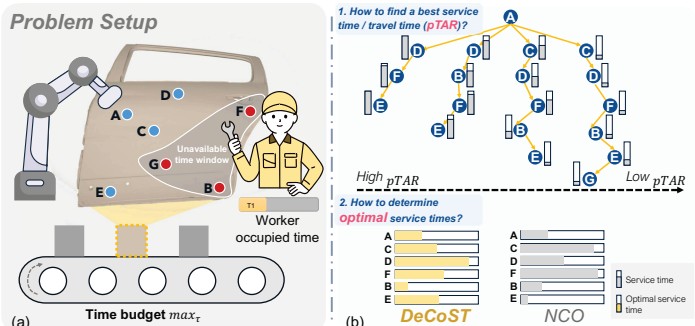

Figure 1: (a) Application of OPTWVP in industrial scenarios. A manipulator and a human are collaborating on a defect removal task on the assembly line. The robot has limited time to operate. It cannot access the nodes outside the time windows for safety reasons, since the human might have collisions with the robot. Meanwhile, the reward depends on the service time spent handling the defect, with the defect size decreasing linearly over time (Wang et al., 2023). (b) Illustration of the challenge faced by NCO methods in allocating hybrid discrete-continuous decisions and capturing continuous-variable representations.

classical VRPs that determine only the routing sequence, OPTWVP requires jointly optimizing the route and the service time spent at each node, because the reward depends directly on service times on vertices. A selected route dynamically shapes the feasible region of service time decisions through travel times and time window constraints, while the allocated service times, in turn, affect the obtained rewards and influence the overall route decisions. This bidirectional dependency prevents the two components from being optimized independently, causing an exponential expansion of the joint search space.

At present, the main strategies for addressing OP/VRP variants can be generally divided into heuristic/metaheuristic methods and learning-based neural combinatorial optimization (NCO) methods. Heuristic and metaheuristic algorithms typically follow a multistage strategy, where an initial feasible solution is first generated and subsequently refined with local search improvements (Nikzad et al., 2023; Pirabán-Ramírez et al., 2022; Canca et al., 2024). Although these approaches demonstrate good performance on specific variants, their performances are often heavily limited by manually designed heuristics and exhaustive refinement procedures. On the other hand, the rapidly growing NCO methods (Kool et al., 2019; Kwon et al., 2020) seek to apply learning-based approaches to increase the search efficiency in classic VRPs. In pursuit of more practical applicability (Bi et al., 2024; Hottung et al., 2025a; Wang et al., 2025), more recent studies in NCO also explore VRP variants with constraints. They adopt problem deconstruction methods (Hottung et al., 2025b; Kuang et al., 2024) to decompose the original problem by predicting a certain number of variables and then performing search/optimization within their local neighborhoods. However, most NCO methods focus solely on routing, while the joint optimization of routing and service-time allocation remains under-explored. In these problems, such decomposition leads to shortsighted route predictions that cannot anticipate optimal service times, and the subsequent local adjustments are insufficient to correct the structural biases introduced in the first stage. Unlike the above unsupervised NCO methods, current Mixed-Integer Programming (MIP) problem studies provide a similar solution to general MIP problems (Liu et al., 2025; Paulus & Krause, 2023; HU et al., 2024), but most of these methods rely on annotated data, which limits their applications. Taken together, for problems like OPTWVP, where discrete routing decisions and continuous service-time allocation are inherently interdependent, achieving an efficient and high-quality solution remains unsolved.

In this paper, we propose DEcoupled discrete-Continuous optimization with Service-time-guided Trajectory (DeCoST), a learning-based two-stage collaborative framework proposed for solving OPTWVP-like problems. Leveraging the structure of OPTWVP, DeCoST decomposes OPTWVP into a routing problem and a service time allocation problem, which effectively decouples the searching process while enabling efficient and learnable coordination. In the first stage, DeCoST employs a parallel decoder that integrates a routing decoder and a service time decoder (STD) to provide a trajectory generation strategy and an initial estimate of service times, which explicitly considers the influence of continuous variables on the quality of the solution and provides an initial coordination between discrete path decisions and continuous service time allocations. In the second stage, based

on the fixed discrete path trajectory, the continuous decision variable optimization problem is further simplified to a linear programming (LP) problem. Especially, a service time optimization (STO) algorithm is applied in the second stage to guarantee parallel computation and maximize the total collected reward. We rigorously prove that the STO can obtain the global optimum. In this two-stage approach, we introduce a repulsive supervisory index, profit-weighted time allocation ratio (pTAR), to provide a feedback signal for service time prediction. By preventing premature convergence to the deterministic conditional optimum, the model preserves flexibility in service-time prediction and improves the overall solution quality.

The overall contributions of this paper are as follows.

(1) We propose DeCoST, a learning-based two-stage approach for solving OPTWVP, where discrete routing decisions and continuous service-time allocations are decomposed, realizing the joint optimization of routing and service time under the constraints of time windows and variable profits.

(2) By leveraging parallel computation of STO and the cross-stage feedback mechanism, the proposed two-stage approach provides a global long-horizon structure estimation in the early route decision stage, enabling us to search the route and service time efficiently.

(3) We conduct thorough experiments demonstrating that the proposed method outperforms not only existing state-of-the-art (SOTA) NCO approaches but also the latest metaheuristic algorithms in both solution quality and computational efficiency.

## 2 RELATED WORK

**Heuristic and Meta-Heuristic Optimization of COPs with Time Constraints.** Traditional heuristic and metaheuristic approaches typically employ a multistage strategy to solve COP variants with varying profits (Marzal & Sebastia, 2024; Wan et al., 2024; Nikzad et al., 2023; Pirabán-Ramírez et al., 2022; Canca et al., 2024; Panadero et al., 2022; Evers et al., 2014). Initially, greedy strategies are used to construct initial heuristic solutions, which are then refined through active search algorithms (Marzal & Sebastia, 2024). Yu et al. (2019) also introduces a multi-phase approach to separately handle discrete and continuous decision variables. Although effective on specific problems, these methods face inherent challenges in solving OPTWVP-like problems: they depend on manually designed heuristics. Moreover, any route change reshapes all nodes' feasible service-time intervals, which affects the overall objective. As a result, local operations cannot be evaluated using local information, and each neighborhood operation requires exhaustively solving continuous subproblems.

**Neural Combinatorial Optimization.** Recent advances in NCO methods have focused primarily on improving solution quality (Kwon et al., 2021; Guo et al., 2024; Zhang et al., 2023b), scalability (Qiu et al., 2022; Ye et al., 2024; Zheng et al., 2024; Luo et al., 2023; Li et al., 2021; Su et al., 2023), and generalization (Kool et al., 2019; Kwon et al., 2020; JIANG et al., 2023; Zhang et al., 2023a; Zhou et al., 2024; Berto et al., 2025a; Wan et al., 2023; Kim et al., 2022; Drakulic et al., 2023; Li et al., 2025a; Bi et al., 2022; Berto et al., 2025b). NCO approaches can be categorized into autoregressive (AR) and non-autoregressive (NAR) methods. AR methods adopt an end-to-end generative approach to solve COPs (Chalumeau et al., 2023). In contrast, NAR methods (Ma et al., 2023) focus on learning structural information directly to guide the construction of solutions (Min et al., 2023; Xin et al., 2021; Sun & Yang, 2023; Li et al., 2024) by generating intermediate representations, which are then used to guide meta-heuristic algorithms (Heydaribeni et al., 2024; Choo et al., 2022). However, these algorithms are not directly applicable because they lack the ability to effectively handle constraints.

**NCO with Constraints.** Recent NCO studies incorporate exploring more realistic and complex constraints in variants of COPs (Huang et al., 2023). For example, time-window constraints have been introduced in COP formulations (Chen et al., 2024; Bi et al., 2024), where multistep estimation strategies are adopted to ensure feasibility. Hypergraph-based approaches are also applied to deal with constrained COPs (Wang et al., 2025). However, most NCO methods focus solely on routing, while the joint optimization of routing and service-time allocation remains under-explored, leading to shortsighted routing that overlooks nodes' interconnection with service times. In MIP settings, coupled discrete-continuous decision variables are dealt with through model reductions approach (Li et al., 2025b), hybrid reinforcement learning (Zhang et al., 2024), contrastive learning (Han et al.,

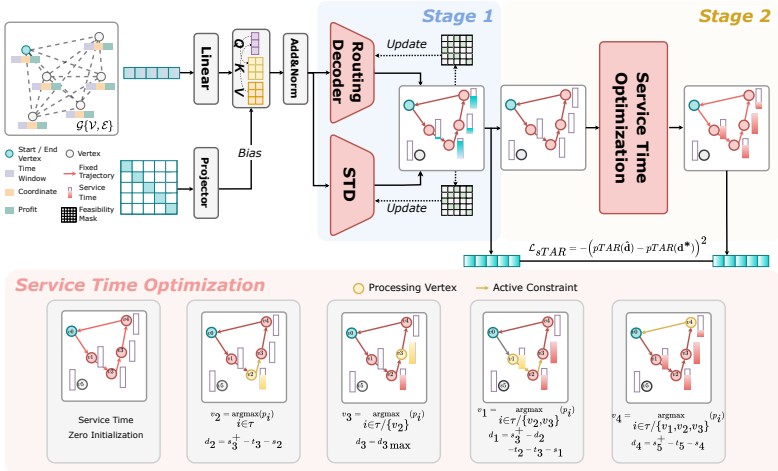

Figure 2: An overview of the DeCoST approach. The upper part illustrates the two-stage collaborative optimization process, while the lower part presents the details of the STO Algorithm 1.

2023; Huang et al., 2024), and so on. However, most of these methods rely on annotated data, which limits their generalization in scenarios without supervised data.

## 3 PRELIMINARIES

An OP instance can be explained on an undirected complete graph $\mathcal{G}\{\mathcal{V}, \mathcal{E}\}$ with $n$ vertices, where $\mathcal{V} = \{v_0, v_1, \cdots, v_{n-1}\}$, $\mathcal{E} = \{e_{i,j} | i, j \in \mathcal{V}\}$. The vertex $v_0$ represents the start depot and the end depot. All vertices $v_i$ and $v_j \in \mathcal{V}$ are connected by the edge $e_{i,j}$, and the corresponding travel time from node $i$ to node $j$ is denoted as $t_{ij}$. $r : \mathcal{V} \mapsto \mathbb{R}$ is the reward function associated with each vertex. The OPTWVP aims to find a path $\tau = \{v_0, v^{(1)}, ..., v^{(l-2)}, v_0\}$ (of length $l$) and the corresponding service time $\mathbf{d} \in \mathbb{R}^l$, where all intermediate vertices $v^{(j)} \in \mathcal{V} \setminus \{v_0\}$ are visited only once, and maximize the total reward $\sum_{v_j \in \tau} r(v_j)$ collected along the path $\tau$, subject to:

(1) Time limit: The total travel and service time must not exceed the given time limit $\max_\tau$;

(2) Service time boundaries: The service time $d_i$ of each visited vertex must satisfy $d_i \in [0, d_{i\,\max}]$;

(3) Time window constraints: The start time $s_i$ of each visited vertex must lie in the time window $[s_i^-, s_i^+]$, i.e., $s_i \in [s_i^-, s_i^+]$;

(4) Time varying profits: Each vertex has a corresponding unit profit $p_i$. The profit of each vertex is related to the service time $d_i$ and the unit profit $p_i$ of the node, which is defined by $f(d_i, p_i) = p_i d_i$.

Several classical OP variants, including the OP, OP with variable profits (OPVP), and OP with time windows (OPTW), are special cases of OPTWVP. See Appendix C for details and mathematical representation of OPTWVP (Marzal & Sebastia, 2024).

## 4 METHODOLOGY

To enable the adoption of RL techniques, we cast the OPTWVP as a constrained Markov Decision Process (CMDP) defined by the tuple $(\mathcal{S}, \mathcal{A}, \mathcal{P}, \mathcal{C}, R)$. $\mathcal{S}$ is the infinite state space and $A$ denotes the infinite action space. Each state $\sigma_i \in S$ encodes the previous actions $\{a_0, a_1, \cdots, a_k\}$, where an action $a_i = (v^{(i)}, d_{v^{(i)}}/d_{v^{(i)}\,\max}) \in \mathcal{A}$ includes a discrete number indicating the selection of the vertex and a continuous number $\delta d_i \in [0, 1] \subset \mathbb{R}$ indicating the normalized service time ratio at the vertex $v_i$, where $\delta d_{v^{(i)}} = d_{v^{(i)}}/d_{v^{(i)}\,\max}$. $\mathcal{P}$ is the transition probability function, $\mathcal{C}(\sigma, a) = [c_1(\sigma, a), \cdots, c_{\kappa n}(\sigma, a)]^T$ is the set of corresponding constraint functions and $R(\tau, \mathbf{d}|\mathcal{G}) = \sum_{i=1}^{l-1} p_{v^{(i)}} d_{v^{(i)}}$ denotes the cumulative reward of the given trajectory $\tau$ and the corre-

sponding service time $\mathbf{d}$, where $\kappa$ denotes the number of constraint functions defined for each vertex. $p_{v^{(i)}}$ is the unit profit of vertex $v^{(i)}$, and $d_{v^{(i)}}$ is the service time.

The objective of CMDP is to learn a policy $\pi_\theta : \mathcal{S} \rightarrow \mathcal{P}(\mathcal{A})$ that maximizes the overall reward:

$$
\begin{aligned}
\max_{\pi_\theta} \quad & J_r(\pi_\theta) = \mathbb{E}_{(\tau,\mathbf{d}) \sim \pi_\theta} \left[ R(\tau, \mathbf{d}|\mathcal{G}) \right], \\
\text{s.t.} \quad & \pi_\theta \in \Pi_c, \Pi_c = \left\{ \pi_\theta | \mathcal{C}(\sigma_i, a_i) \leq \mathbf{0}, \forall i \in N \right\},
\end{aligned}
\tag{1}
$$

where $J_r$ is the expected return, and $\Pi_c$ denotes the set of feasible policies satisfying all OPTWVP constraints. At each step $i$, the action is sampled from the policy $\pi_\theta$. We denote this sampling process compactly as $(\tau, \mathbf{d}) \sim \pi_\theta$, where both the path and service allocation are jointly generated by the policy. The overall policy $\pi_\theta$ is trained by REINFORCE (Williams, 1992) using the following loss function:

$$
\mathcal{L} = -\frac{1}{M} \sum_{i=1}^{M} (R(\tau, \mathbf{d}|\mathcal{G}) - b) \log(\pi_\theta),
\tag{2}
$$

where $b = \frac{1}{M} \sum_{i=1}^{M} R(\tau, \mathbf{d}|\mathcal{G})$ denotes the average reward of the trajectories among $M$ batch samples.

To address the inherent interdependency between discrete routing decisions and continuous service-time allocation, we propose a two-stage DeCoST framework to decompose and coordinate the path selection and service time allocation. As shown in Figure 2, in the first stage, DeCoST employs a parallel decoder that integrates a routing decoder and an STD to provide a path generation strategy and an initial estimation of service times, which jointly generate the discrete–continuous action pair $a_i$ used during path construction. In the second stage, based on the fixed discrete path trajectory, the service time allocation problem is simplified into an LP problem, which is efficiently solved by STO. Given the initial service times $\hat{\mathbf{d}}$ in the first stage and the optimal service time $\mathbf{d}^*$ in the second stage, pTAR loss of $\hat{\mathbf{d}}$ and $\mathbf{d}^*$ are calculated to encourage STD to broadly explore policies, thereby enabling an early and reliable estimation of problem structure and improving the first-stage service-time prediction. The following sections introduce the specific design of the two-stage optimization process and the service time supervision mechanism.

## 4.1 Two-stage optimization

**Determine the feasible trajectory.** In the first stage, the policy $\pi_\theta$ uses a parallel decoder structure with a route decoder and an STD. The route decoder selects the next node to visit, while the STD simultaneously predicts the corresponding service time. Together, they generate a feasible trajectory $\tau$ along with the corresponding service times $\hat{\mathbf{d}}$.

Two techniques are integrated into the generation process to improve the quality of the constructed solutions and ensure their feasibility: spatial encoding for structural awareness and feasibility masking for time window constraints. These enhancements enable the model to capture graph connectivity better while avoiding infeasible decisions during trajectory construction.

Spatial encoding (Ying et al., 2021) is applied to the solver to incorporate edge features. Therefore, the solver not only encodes node features, but also encodes edge features (node distances) as an attention bias, thereby improving the model's understanding of the graph structures and enabling the model to pay more attention to node combinations with lower costs and denser connections.

In addition, similar to Bi et al. (2024), a feasibility masking is applied to guarantee the feasibility of the generated trajectory: the mask dynamically excludes candidate vertices that would violate constraints in one of the following two cases: i) When adding a new vertex and returning to $v_0$ within the time limit $max_\tau$ is impossible. ii) When the start time of the next vertex exceeds its time window. By incorporating feasibility masking, the model is guided to construct only feasible trajectories in the first stage, which significantly reduces the search space.

The feasible trajectory $\tau$ is subsequently fixed in the second stage, allowing the decoupling of the discrete routing variables and the continuous service times.

**Service time optimization.** Given the fixed trajectory, in the second stage, OPTWVP is decomposed into a decoupled continuous LP problem that maximizes the total collected rewards. The second

---
**Algorithm 1** Service time optimization algorithm

---
**Input:** Trajectory $\tau$, distance matrix $\mathbf{t}$, time windows $(\mathbf{s}^-, \mathbf{s}^+)$, max service time $\mathbf{d}_{\max}$, limited time $\max_\tau$.
**Output:** Optimal service time allocation $\mathbf{d}^*$, optimal total reward $\mathbf{r}^*$
1: Initialize: trajectory length $l$, service duration $\mathbf{d} \leftarrow \mathbf{0}$, start time $\mathbf{s} \leftarrow \mathbf{0}$
2: $d_0 \leftarrow 0, s_0 \leftarrow 0, s_1 \leftarrow \max\{s_1^-, t_1\}$ // initialization
3: **for** $i = 0$ to $l - 2$ **do**
4: $\quad s_{i+1} \leftarrow \max\{s_i + t_{i+1}, s_{i+1}^-\}$ // start time update
5: $sorted \leftarrow$ From high to low, sorted indices of nodes in $\tau$.
6: **for all** $i$ in $sorted$ **do**
7: $\quad d_i = \min\{d_{i\,\max}, \min_{k=0}^{n-i-1}\{s_{i+k+1}^+ - s_i - \sum_{j=1}^k d_{i+j} - \sum_{j=1}^{k+1} t_{i+j}\}\}$
8: $\quad$ **for** $j = i + 1$ to $n - 1$ **do**
9: $\quad\quad s_j \leftarrow \max\{s_{j-1} + d_{j-1} + t_j, s_j^-\}$ // start time update
10: $\mathbf{d}^* \leftarrow \mathbf{d}, \mathbf{s}^* \leftarrow \mathbf{s}, R^* \leftarrow \mathbf{p}^T \mathbf{d}^*$

---

stage service time optimization problem, expressed in matrix form, is given as follows:

$$
\begin{aligned}
\max_{\mathbf{s},\mathbf{d}} \quad & \mathbf{p}^T \mathbf{d}, \\
\text{s.t.} \quad & \mathbf{s}^- \leq \mathbf{s} \leq \mathbf{s}^+, \\
& \mathbf{0} \leq \mathbf{d} \leq \mathbf{d}_{\max}, \\
& (I - U)\mathbf{s} + \mathbf{d} + \mathbf{t} \leq \mathbf{0},
\end{aligned}
\tag{3}
$$

where $\mathbf{p} = [p_0, \ldots, p_{l-1}]^\top$, $\mathbf{d} = [d_0, \ldots, d_{l-1}]^\top$, $\mathbf{d}_{\max} = [d_{1\,\max}, \ldots, d_{l-1\,\max}]$, $\mathbf{s} = [s_0, \ldots, s_{l-1}]^\top$, $\mathbf{s}^- = [s_0^-, \ldots, s_{l-1}^-]^\top$, $\mathbf{s}^+ = [s_0^+, \ldots, s_{l-1}^+]^\top$, $\mathbf{t} = [t_0, \ldots, t_{l-1}] \in \mathbb{R}^l$ respectively denote profit, service time, maximum service time, start time, start, end of time window and travel time in the selected trajectory $\tau$. $I$ and $U \in \mathbb{R}^{l \times l}$ are the identity matrix and the upper shift matrix, and $l$ denotes the trajectory length.

An STO algorithm is introduced that performs parallel computation to obtain the optimal service-time allocation, as shown in Algorithm 1. The algorithm first constructs a feasible solution, then iteratively selects the vertex with the highest reward and assigns a service time that satisfies both the end of time-window constraints and the maximum service-time limit. The start times of subsequent nodes are updated accordingly. The procedure terminates once the overall time budget $\max_\tau$ is reached.

Theorem 4.1 hereafter states the optimality of the solution computed by Algorithm 1.

**Theorem 4.1.** *The service time optimization algorithm shown in Algorithm 1 returns an optimal solution $(s^*, d^*)$ to the service time scheduling problem specified in Equation (3).*

The proof of this theorem can be found in Appendix A.

## 4.2 SUPERVISED LEARNING OF SERVICE TIME DECODER

Although the trajectory $\tau$ and initial service time $\hat{\mathbf{d}}$ have been constructed through the constructive solver, and a set of optimal service times have been further calculated, the quality of initial service time allocation has a decisive influence on the quality of the final solution. This results from a significant trade-off in how the total time budget is allocated between travel and service. Improper service time allocation may lead to two extreme situations: (1) *Low service time*: The model prefers to access more vertices, but each node is underserved, resulting in a potential low reward; (2) *High service time*: Although the reward of a single vertex is increased, more optional nodes are sacrificed, which might reduce the overall reward.

This indicates that inappropriate travel-service time allocation can critically damage the quality of the solution. To learn the trade-off between travel time and service time allocation and better guide the policy of the initial service time assignment, a learnable metric, named the profit-weighted time allocation ratio (pTAR), is introduced. It is defined by the sum of profit-weighted service times $p_i d_i$ versus travel times $t_i$ of the vertices in the trajectory $\tau$:

$$
pTAR(\mathbf{d}) = \sum_{i \in \tau} \frac{p_i d_i}{t_i}.
\tag{4}
$$

The pTAR metric reflects the profit efficiency of a solution, measuring how much total reward is achieved per unit of travel cost. A higher pTAR value indicates a strategy that focuses more on rewarding regions with lower travel expense, which correlates with the quality of the constructed trajectory. To prevent the model from overfitting to the LP-derived conditional optimum, a repulsive supervisory loss is further defined as:

$$\mathcal{L}_{pTAR} = -(pTAR(\hat{\mathbf{d}}) - pTAR(\mathbf{d}^*))^2, \tag{5}$$

REINFORCE (Williams, 1992) trains the policy $\pi_\theta$, and the overall loss function is designed as a summation of REINFORCE loss and the supervisory loss of service time, expressed as:

$$\mathcal{L}_{total} = \beta_1 \mathcal{L} + \beta_2 \mathcal{L}_{pTAR}, \tag{6}$$

where $\beta_1, \beta_2$ denotes the tuning parameter.

## 5 EXPERIMENTS

**Implementation details.** We conduct experiments on the OPTWVP benchmark, a typical VRP with continuous time constraints. The methods are tested on the datasets with different time window sizes (TW = 100 and TW = 500) and node numbers ($n = 50$, $n = 100$, and $n = 500$). The training set has 10000 instances for each dataset, and the test set has 10000 instances for $n = 50$ TW = 100, $n = 100$ TW = 100, and 100 instances for the others. The network is trained for 800 epochs, with an early stop strategy of stopping when there is no improvement for 100 epochs. We implement the proposed approach in Python using the PyTorch library. All runtime experiments are performed on a laptop with NVIDIA GeForce RTX 4070, Intel (R) Core(TM) i7 CPU 2.20GHz to ensure consistency, while all score experiments are performed on both the laptop and a server with NVIDIA GeForce RTX 5090 GPU, AMD Ryzen 9 9950X 16-Core Processor to confirm robustness across platforms. Please refer to Appendix E for more details on the implementation. The code is publicly available at https://github.com/SwonGao/DeCoST.

**Baselines.** The proposed approach is compared with three types of baselines: 1) Commercial optimization tool: Gurobi (Gurobi Optimization, LLC, 2024), which implements the branch and cut (B&C) algorithm to obtain the exact solution. 2) Meta-heuristic algorithms: Greedy-Profit ratio search (Greedy-PRS), a two-stage greedy heuristic that first determines a route by selecting nodes based on the highest unit profit and then optimizes service times using PuLP (Mitchell et al., 2011) without altering the route; Incremental local search (ILS) (Marzal & Sebastia, 2024), a metaheuristic search algorithm that implements fast first solution and local search. 3) Learning-based solvers: Since there are no current NCO methods to solve OPTWVP, we implemented greedy methods and our two-stage approach to deal with service time in POMO (Kwon et al., 2020) and GFACS (Kim et al., 2025).

**Evaluation metrics.** Three metrics are adopted to evaluate the performance of the methods: 1) Score: the average total collected rewards of the solutions within the test sets. 2) Gap: $Gap = (R^* - R)/R^* \times 100\%$ denotes the average optimality gap between the optimal reward $R^*$ w.r.t. the strong baseline Gurobi (Gurobi Optimization, LLC, 2024) and the reward $R$ obtained within the test sets. Gurobi is set to run until the global optimum is proven, ensuring that the computed $R^*$ corresponds to the exact optimal solution. 3) Runtime: The average runtime denotes the average inference time required to find a solution for the instance within the test sets. For the ILS method, the reported runtime corresponds to the time to reach the reported score within the search time limit, which is set to 10 seconds.

### 5.1 PERFORMANCE ON SOLVING OPs WITH TIME CONSTRAINTS

To comprehensively evaluate the performance of the DeCoST framework on the extended OPTWVP problem, we compare it against several representative baselines, with results summarized in Table 1. The results show that DeCoST consistently achieves strong performance in all settings. It outperforms other heuristic and NCO methods in terms of solution quality (Score and Gap), while maintaining high computational efficiency. Adding the STO module significantly improves the NCO methods. For example, GFACS with STO reduces the gap from 13.4% to 3.38% in the setting $n = 100$, TW = 100. The increased runtime (up to 6760ms) is mainly due to the non-autoregressive structure

Table 1: Performance comparison on OPTWVP under different problem sizes and time window settings. **Bold** values denote the best performance in each setting, and underlined values denote the second-best. Category **C** refers to NCO methods, and **H** refers to heuristic baselines. The default unit for runtime is milliseconds (ms).

| Cat. | Method | $n = 50$, TW $= 100$ | | | $n = 100$, TW $= 100$ | | |
|---|---|---|---|---|---|---|---|
| | | Score ↑ | Gap ↓ | Runtime ↓ | Score ↑ | Gap ↓ | Runtime ↓ |
| – | B&C | 15.2 | 0.00% | 200 | 26.1 | 0.00% | 1023 |
| H | Greedy-PRS | 12.9 | 15.7% | 44 | 21.9 | 16.2% | 48 |
| | ILS | 14.5 | 4.34% | 2109 | 24.9 | 4.2% | 7122 |
| C | GFACS (Greedy) | 11.1 | 25.7% | 94.2 | 22.6 | 13.4% | 98.2 |
| | GFACS | 12.3 | 18.6% | 4310 | 25.2 | 3.38% | 6760 |
| | POMO | 11.3 | 25.3% | 46.6 | 11.5 | 55.7% | 52.3 |
| | **DeCoST (Ours)** | **15.1** | **1.06%** | 94.0 | **25.6** | **1.97%** | 158 |
| Cat. | Method | $n = 50$, TW $= 500$ | | | $n = 100$, TW $= 500$ | | |
| | | Score ↑ | Gap ↓ | Runtime ↓ | Score ↑ | Gap ↓ | Runtime ↓ |
| – | B&C | 51.9 | 0.00% | 1346s | – | – | >24h |
| H | Greedy-PRS | 46.1 | 11.4% | 46.1 | 79.7 | – | 51 |
| | ILS | 47.8 | 7.82% | 1262 | 85.0 | – | 8441 |
| C | GFACS (Greedy) | 35.21 | 31.7% | 89.8 | 60.5 | – | 97.8 |
| | GFACS | 47.4 | 8.57% | 6950 | 84.4 | – | 8750 |
| | POMO | 31.3 | 39.6% | 40.7 | 54.6 | – | 60.6 |
| | **DeCoST (Ours)** | **51.4** | **0.83%** | 60.4 | **88.1** | – | 100 |

Table 2: Performance on OPTWVP with large-scale configuration ($n = 500$, TW $= 100$). **Bold** values indicate the best result.

| Method | Score ↑ | Gap ↓ | Runtime (ms) ↓ |
|---|---|---|---|
| B&C | 82.3 | 0.00% | 68400 |
| Greedy-PRS | 69.6 | 15.6% | 60 |
| ILS | 78.2 | 4.98% | 8803 |
| GFACS (Greedy) | 67.4 | 18.1% | 112 |
| GFACS | 73.1 | 11.3% | 9420 |
| POMO | 58.6 | 28.8% | 747 |
| **DeCoST (Ours)** | **79.6** | **3.31%** | 1329 |

of GFACS, which limits the batch parallelism of STO and forces more frequent optimization of single instances. Although runtime rises, the substantial increase in solution quality demonstrates that precise optimization of service time is crucial for achieving a better solution. To ensure a fair comparison, the STO module is only integrated into NCO frameworks, which lack native mechanisms for service time allocation. We do not apply STO to heuristic baselines like ILS or Greedy-PRS, as they already incorporate built-in optimization strategies. Compared to strong meta-heuristic baselines like ILS, DeCoST yields better or comparable solution quality and improves inference efficiency by a factor of 20 to 45. We also include supplementary experiments on comparison with ILS with search time limit >10s in Appendix B.1.

DeCoST also demonstrates strong performance and scalability in solving large-scale OPTWVP instances. For instance, with $n = 500$, as shown in Table 2, DeCoST maintains high solution quality while requiring only 1329ms on average—significantly faster than ILS, which takes 8803ms. Although a slight degradation in solution quality is expected as the problem size increases—a common trend in combinatorial optimization, DeCoST consistently outperforms all competing baselines.

Figure 3 reveals the distribution of the optimality gap of different methods under different settings (Refer to Appendix B.2 for more details). It can be observed that DeCoST shows the smallest optimality gap in all settings, with a low box height and fewer outliers, indicating that the method always maintains high solution quality and consistency on multiple instances and has significant stability.

Finally, we also include extension experiments on Team OPTWVP, sensitivity analysis with respect to cost function parameters, performance evaluation on initial service times, efficiency study of pTAR loss in Appendix B.3, B.4, B.5, and B.6, respectively.

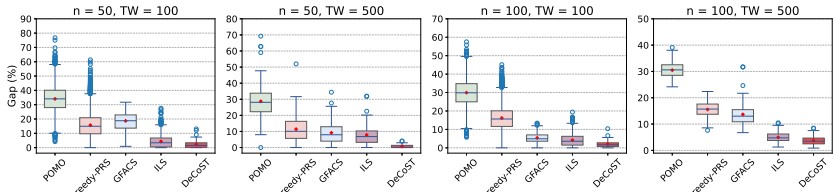

Figure 3: Boxplot of the optimality gap under different problem sizes and time window settings.

Table 3: Ablation study on DeCoST ($n = 50$). **Bold** numbers denote the best in each block (TW).

| Method / Setting | SE | STO | SL | Score ↑ | Gap ↓ | Runtime (ms) ↓ |
|---|---|---|---|---|---|---|
| **TW = 100** | | | | | | |
| Baseline | ✗ | ✗ | ✗ | 11.3 | 25.3% | 46.6 |
| Baseline + SE | ✓ | ✗ | ✗ | 11.6 | 23.0% | 46.7 |
| Baseline + SE + STO | ✓ | ✓ | ✗ | 14.9 | 2.28% | 77.5 |
| **DeCoST (ours)** | ✓ | ✓ | ✓ | **15.1** | **1.06%** | 94.0 |
| **TW = 500** | | | | | | |
| Baseline | ✗ | ✗ | ✗ | 31.3 | 39.6% | 40.7 |
| Baseline + SE | ✓ | ✗ | ✗ | 35.5 | 31.2% | 39.9 |
| Baseline + SE + STO | ✓ | ✓ | ✗ | 50.9 | 1.88% | 55.6 |
| **DeCoST (ours)** | ✓ | ✓ | ✓ | **51.4** | **0.83%** | 60.4 |

## 5.2 PERFORMANCE ON DECOST WITH MODULE ABLATIONS

Table 3 shows the ablation results of the three components in the DeCoST framework: the spatial encoder (SE) that incorporates node-edge features, the STO module, and the supervised loss (SL) guided by our proposed pTAR metric. Experiments are conducted on two time window settings. As shown in the results, the baseline model (without any of the three modules) performs the worst, with Gap values reaching 25.3% and 39.6%. Introducing the SE module yields moderate performance gains without noticeable runtime overhead, indicating the positive impact of geometric structure encoding on routing decisions. However, the most substantial improvement comes from the STO module, which significantly reduces the Gap to 2.28% and 1.88% under the two settings, with only about 30ms of additional runtime. When further augmented with the SL module, the Gap decreases to 1.06% and 0.83% in the TW = 500 setting, demonstrating that supervised guidance from pTAR can further refine the allocation of service time and improve the quality of the solution. In summary, STO plays a decisive role in improving performance, while SE and SL provide complementary enhancements. The synergy among the three components enables DeCoST to deliver near-optimal solution quality with minimal inference time, validating the effectiveness and coherence of the overall design.

## 5.3 PERFORMANCE ON SOLOMON100

We also evaluated DeCoST with ILS on the data set Solomon100[1], a widely used standard dataset in the vehicle routing problem, with the same configuration of (Marzal & Sebastia, 2024). The Solomon100 dataset consists of 101-node TOPTWVP instances. Due to the limited number of available samples, we select 29 instances for training and 10 instances for testing, with a training schedule of 1000 epochs. The training and validation batch sizes are set to 2 and 1, respectively. The performance is summarized in Table 4. DeCoST consistently outperforms ILS on this real-world dataset under both 10 s and

Table 4: Performance comparison on OPTWVP on Solomon 100.

| Method | Score ↑ | Gap ↓ |
|---|---|---|
| B&C | - | - |
| ILS (T=10s) | 9846.3 | -7.8% |
| ILS (T=20s) | 10616.9 | -0.59% |
| DeCoST (T=0.29s) | **10680.3** | – |

20 s time limits. Notably, DeCoST achieves high-quality solutions at approximately 34.5 times faster inference speed compared to ILS.

## 6 CONCLUSION AND FUTURE WORK

This paper proposes a two-stage DeCoST framework to explicitly decouple discrete and continuous decision variables for the OPTWVP: the first stage fixes the routing trajectory and the second stage

---

[1]https://www.sintef.no/projectweb/top/vrptw/solomon-benchmark/

optimizes the service times through STO. A mathematical proof demonstrates the global optimality of the second-stage solution. Furthermore, a repulsive supervisory metric, pTAR, incorporates optimal service times into the first-stage trajectory decision to learn a global long-horizon structure estimation, thereby improving the overall quality and efficiency of the solution. Experiments on OPTWVP instances demonstrate that the proposed method outperforms both state-of-the-art NCO methods and the latest meta-heuristic algorithms in terms of solution quality and computational efficiency. Although our method demonstrates strong accuracy in both AR and NAR solvers, a limitation arises when integrating with NAR approaches: the lack of batch processing limits STO's parallelism, thereby impacting overall efficiency. Future work will focus on improving computational efficiency in such settings and extending the framework to more general vehicle routing scenarios.

ACKNOWLEDGMENTS

Co-funded by the European Union under the EU - HE Magician – Grant Agreement 101120731. Views and opinions expressed are however those of the author(s) only and do not necessarily reflect those of the European Union or the European Commission. Neither the European Union nor the granting authority can be held responsible for them.

ETHICS STATEMENT

This work complies with the ICLR Code of Ethics. We did not involve human subjects or personal data. All datasets are publicly available and used under their original licenses. For more details about the datasets, please refer to Appendix E. Our study does not target sensitive attributes and poses no foreseeable risks beyond the usual limitations of machine learning research. We have no conflicts of interest or external sponsorship to declare. For transparency and reproducibility, we provide an anonymized code repository with all configuration files.

REPRODUCIBILITY STATEMENT

All datasets used are publicly accessible, enabling full reproduction of our results. We report all training and evaluation details and hyperparameters in the paper. We also provide an code repository at `https://github.com/SwonGao/DeCoST`.

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

## LLM USAGE DISCLOSURE

We use LLM as a writing aid for grammar checking and for polishing the Introduction. The LLM did not generate technical content, experiments, or results. All claims, data, equations, and code were authored, verified, and are the responsibility of the human authors. No confidential, proprietary, or identifying information was shared with the LLM.

## A   PROOF OF THEOREM 4.1

In this paper, we propose a service time optimization algorithm to obtain an optimal solution for the second-stage service time allocation. The details of the proof are given by:

*Proof.* Consider the Lagrangian of the problem (3) in the matrix form:

$$\mathscr{L}(\mathbf{s}, \mathbf{d}, \lambda) = - \begin{bmatrix} \mathbf{0} & \mathbf{p}^T \end{bmatrix} \begin{bmatrix} \mathbf{s} \\ \mathbf{d} \end{bmatrix} + \begin{bmatrix} \lambda_1^T & \lambda_2^T & \lambda_3^T & \lambda_4^T & \lambda_5^T \end{bmatrix} \begin{bmatrix} \tau^- - \mathbf{s} \\ \mathbf{s} - \tau^+ \\ -\mathbf{d} \\ \mathbf{d} - \tau^+ + \tau^- \\ (I - U)\mathbf{s} + \mathbf{d} + \mathbf{t} \end{bmatrix}, \quad (7)$$

where $\mathbf{s}, \mathbf{d}$ are the start, service time of the given trajectory $\tau$ from Algorithm 1, $U$ denotes the upper shift matrix, and $\lambda = [\lambda_1, \lambda_2, \lambda_3, \lambda_4, \lambda_5]^T$ denotes the Lagrangian multipliers of the corresponding constraints.

Since problem (3) is a constrained convex optimization problem, as long as there exists a solution $(\mathbf{s}, \mathbf{d}, \lambda)$ such that satisfies the Karush–Kuhn–Tucker (KKT) conditions, the solution $(\mathbf{s}, \mathbf{d}), \lambda$ are the optimal solution of the primal and dual problem, respectively Boyd & Vandenberghe (2004).

The KKT conditions of problem (3) can be expressed as:

$$\begin{aligned} s_i^- \leq s_i \leq s_i^+, \\ 0 \leq d_i \leq d_{i\max}, \\ s_i + d_i + t_i - s_{i+1} \leq 0, \\ \lambda_{1,i}(\tau_i^- - s_i) = 0 \\ \lambda_{2,i}(s_i - \tau_i^+) = 0 \\ \lambda_{3,i}(-d_i) = 0 \\ \lambda_{4,i}(d_i - d_{i\max}) = 0 \\ \lambda_{5,i}(s_i + d_i + t_i - s_{i+1}) = 0 \\ -\lambda_{1,i} + \lambda_{2,i} + \lambda_{5,i} - \lambda_{5,i-1} = 0 \\ -\lambda_{3,i} + \lambda_{4,i} + \lambda_{5,i} = p_i \end{aligned} \quad (8)$$

Consider an arbitrary solution $\mathbf{s}, \mathbf{d}$ from algorithm 1; it is a feasible solution $\mathbf{s}, \mathbf{d}$ for the primal problem (3). The optimality of the solution can be guaranteed by finding a feasible solution $\lambda$ for the dual problem.

According to the line 9 of the algorithm, the selected nodes in the route $\tau$ can be separated into three categories: $\overline{\mathcal{V}} = \{v_i | d_i = d_{i\max} = s^+ - s^-, v_i \in \tau\}$, $\hat{\mathcal{V}} = \{v_i | d_i = s_{i+l+1}^+ - s_i - \sum_{j=1}^{l} d_{i+j} - \sum_{j=1}^{l+1} t_{i+j}, v_i \in \tau\}$, and $\underline{\mathcal{V}} = \{v_i | d_i = 0, v_i \in \tau\}$.

For $i \in \overline{V}$, $v_i$ is not constrained by the time windows of other vertices, according to the complementary slackness and gradient condition:

$$\begin{aligned} -\lambda_{1,i} + \lambda_{2,i} + \lambda_{5,i} = \lambda_{5,i-1}, \\ \lambda_{3,i} = 0, \\ \lambda_{4,i} + \lambda_{5,i} = p_i, \end{aligned} \quad (9)$$

For $i \in \hat{V}$, assume that the $d_i$ is constrained by the end of the $i + l + 1$ th node's time window, i.e.

$$d_i = \tau_{i+l+1}^+ - s_i - \sum_{j=1}^{l} d_{i+j} - \sum_{j=1}^{l+1} t_{i+j}, \tag{10}$$

which indicates that the route from $i$ to $i + l$ satisfies:

$$s_k + d_k + t_k = s_{k+1}, k = i, i+1, \cdots, i+l,$$
$$s_{i+l+1} = \tau_{i+l+1}^+, \tag{11}$$

which indicates that the $i$ th vertex to the $i + l$ th vertex are constrained by the end time of $i + l + 1$ th vertex's time window. The corresponding gradient condition and complementary slackness from the $i$ th vertex to the $i + l + 1$ th vertex:

$$\lambda_{5k} = p_k + \lambda_{3k} - \lambda_{4k},$$
$$\lambda_{5k} = \lambda_{1k} - \lambda_{2k} + \lambda_{5k-1}, k = i, \cdots, i+l \tag{12}$$
$$\lambda_{1i+l+1} = 0.$$

For $i \in \underline{V}$, according to the gradient condition and complementary slackness:

$$\lambda_{4i} = 0,$$
$$\lambda_{5i} = p_i + \lambda_{3i}, \tag{13}$$
$$\lambda_{5i} = \lambda_{1i} - \lambda_{2i} + \lambda_{5i-1}.$$

$\forall i \in \overline{V}, \hat{V}, \underline{V}$, the feasibility of the corresponding Lagrange multiplier $(\lambda_{1i}, \lambda_{2i}, \lambda_{3i}, \lambda_{4i}, \lambda_{5i})$ relies solely on the value of $\lambda_{5i-1}$ and satisfies (9), (12), and (13), respectively. So, in the following, different sub-trajectories are constructed to evaluate the feasibility of different sub-trajectories.

Given that each node $i \in \overline{V}, \hat{V}, \underline{V}$ corresponds to a Lagrange multiplier tuple $(\lambda_{1i}, \lambda_{2i}, \lambda_{3i}, \lambda_{4i}, \lambda_{5i})$ whose feasibility depends solely on the value of $\lambda_{5,i-1}$ and satisfies conditions (9), (12), and (13), respectively, the trajectory $\tau$ can be categorized into four types of sub-trajectories. As the feasibility of each node can be recursively determined from the preceding node via $\lambda_{5,i-1}$, it follows that: If each sub-trajectory is locally feasible with respect to its starting value $\lambda_{5,i-1}$, then the entire trajectory $\tau$ is globally feasible.

In the following sections, the feasibility of all possible sub-trajectories is validated by finding a feasible solution that holds sequentially under the recursive propagation of $\lambda_5$.

**Case 1:** $\{V/\overline{V}\} \to \overline{V}$   Consider the case where a sub-trajectory $[i, i+l] \subseteq \{V/\overline{V}\}$ is followed by node $i + l + 1 \in \overline{V}$, KKT conditions yields:

$$\lambda_{5k} = p_k + \lambda_{3k} - \lambda_{4k},$$
$$\lambda_{5k} = \lambda_{1k} - \lambda_{2k} + \lambda_{5k-1}, k = i, \cdots, i+l,$$
$$-\lambda_{1,i+l+1} + \lambda_{2,i+l+1} + \lambda_{5,i+l+1} = \lambda_{5,i+l}, \tag{14}$$
$$\lambda_{3,i+l+1} = 0,$$
$$\lambda_{4,i+l+1} + \lambda_{5,i+l+1} = p_i.$$

$\forall \lambda_{5i-1} \in [0, p_{i+l+1}], \exists (\lambda_{1k}, \lambda_{2k}, \lambda_{3k}, \lambda_{4k}, \lambda_{5k}), k \in [i, i+l+1]$ is the feasible solution of sub-trajectory from $i$ to $i + l + 1$:

$$\lambda_{1,k} = \lambda_{2,k} = \lambda_{4,k} = 0,$$
$$\lambda_{3,k} = p_i - p_k + \lambda_{3,i},$$
$$\lambda_{5,k} = \lambda_{5,k-1}, k = i, \cdots, i+l, \tag{15}$$
$$\lambda_{1,i+l+1} = \lambda_{2,i+l+1} = \lambda_{3,i+l+1},$$
$$\lambda_{5,i+l+1} = \lambda_{5,i+l},$$
$$\lambda_{4,i+l+1} = p_{i+l+1} - \lambda_{5,i-1},$$

such that it satisfies the KKT condition.

**Case 2:** $\{V/\overline{V}\} \rightarrow \{V/\overline{V}\}$   Consider the case where a sub-trajectory $[i, i + l_1] \subseteq \{V/\overline{V}\}$ is followed by another sub-trajectory $[i + l_1 + 1, i + l_1 + l_2 + 1] \subseteq \{V/\overline{V}\}$, KKT conditions yields:

$$
\begin{aligned}
\lambda_{5k} &= p_k + \lambda_{3k} - \lambda_{4k}, \\
\lambda_{5k} &= \lambda_{1k} - \lambda_{2k} + \lambda_{5k-1}, k = i, \cdots, i + l_1 + l_2, \\
\lambda_{1i+l_1+1} &= 0.
\end{aligned}
\tag{16}
$$

$\forall \lambda_{5i-1} \geq 0$, there exists a feasible solution $(\lambda_{1k}, \lambda_{2k}, \lambda_{3k}, \lambda_{4k}, \lambda_{5k}), k \in [i, i + l_1 + l_2 + 1]$ of sub-trajectory from $i$ to $i + l_1 + l_2 + 1$, such that:

$$
\begin{aligned}
\lambda_{1k} &= \lambda_{2k} = \lambda_{4k} = 0, \\
\lambda_{3k} &= p_i - p_k + \lambda_{3i}, \\
\lambda_{5k} &= \lambda_{5k-1}, k = i, \cdots, i + l_1 + l_2,
\end{aligned}
\tag{17}
$$

which satisfies the KKT condition.

**Case 3:** $\overline{V} \rightarrow \overline{V}$   Consider the case where a node $i \in \overline{V}$ is followed by node $i + 1 \in \overline{V}$, KKT conditions yields:

$$
\begin{aligned}
-\lambda_{1,i} + \lambda_{2,i} + \lambda_{5,i} &= \lambda_{5,i-1}, \\
\lambda_{3,i} &= 0, \\
\lambda_{4,i} + \lambda_{5,i} &= p_i, \\
-\lambda_{1,i+1} + \lambda_{2,i+1} + \lambda_{5,i+1} &= \lambda_{5,i}, \\
\lambda_{3,i+1} &= 0, \\
\lambda_{4,i+1} + \lambda_{5,i+1} &= p_{i+1},
\end{aligned}
\tag{18}
$$

$\forall \lambda_{5i-1} \in [0, \min\{p_i, p_{i+1}\}]$, there exists a feasible solution $(\lambda_{1k}, \lambda_{2k}, \lambda_{3k}, \lambda_{4k}, \lambda_{5k}), k \in [i, i + 1]$ of sub-trajectory from $i$ to $i + l + 1$, such that:

$$
\begin{aligned}
\lambda_{1,i} &= \lambda_{2,i} = \lambda_{3,i} = 0, \\
\lambda_{5,i} &= \lambda_{5,i-1}, \\
\lambda_{4,i} &= p_i - \lambda_{5,i-1}, \\
\lambda_{1,i+1} &= \lambda_{2,i+1} = \lambda_{3,i+1} = 0, \\
\lambda_{5,i+1} &= \lambda_{5,i-1}, \\
\lambda_{4,i+1} &= p_{i+1} - \lambda_{5,i-1},
\end{aligned}
\tag{19}
$$

which satisfies the KKT condition.

**Case 4:** $\overline{V} \rightarrow \{V/\overline{V}\}$   Consider the case where a node $i \in \overline{V}$ is followed by a sub-trajectory $[i + 1, i + l + 1] \subseteq \{V/\overline{V}\}$, KKT conditions yields:

$$
\begin{aligned}
-\lambda_{1,i} + \lambda_{2,i} + \lambda_{5,i} &= \lambda_{5,i-1}, \\
\lambda_{3,i} &= 0, \\
\lambda_{4,i} + \lambda_{5,i} &= p_i, \\
-\lambda_{3,k} + \lambda_{4,k} + \lambda_{5,k} &= p_k, \\
\lambda_{5,k} - \lambda_{1,k} + \lambda_{2,k} &= \lambda_{5,k-1}, k \in [i + 1, i + l + 1].
\end{aligned}
\tag{20}
$$

$\forall \lambda_{5i-1} \in [0, p_i]$, there exists a feasible solution $(\lambda_{1k}, \lambda_{2k}, \lambda_{3k}, \lambda_{4k}, \lambda_{5k}), k \in [i, i + l + 1]$ of sub-trajectory from $i$ to $i + l + 1$ such that:

$$
\begin{aligned}
\lambda_{1,i} &= \lambda_{2,i} = \lambda_{3,i} = 0 \\
\lambda_{5,i} &= \lambda_{5,i-1}, \\
\lambda_{4,i} &= p_i - \lambda_{5,i}, \\
\lambda_{1,k} &= \lambda_{2,k} = \lambda_{4,k} = 0, \\
\lambda_{3,k} &= p_i - p_k + \lambda_{3,i}, \\
\lambda_{5,k} &= \lambda_{5,k-1}, k \in [i, i + l + 1],
\end{aligned}
\tag{21}
$$

which satisfies the KKT condition.

To this end, given an arbitrary trajectory $\tau$ and the solution $(s, d)$ from algorithm 1, $\forall \lambda_{5k} \in \min_{i \in [0,l]}\{p_i\}, k \in [0, l]$, there exists a feasible $\lambda$ such that $(s, d, \lambda)$ satisfies the KKT conditions, which demonstrates the optimality of the primal and dual problems. $\qquad \square$

# B    SUPPLEMENTARY EXPERIMENTS

## B.1    RESULTS ON RUNNING THE ILS FOR LONGER THAN 10 SECONDS

To ensure a stronger comparison between ILS and DeCoST, we additionally evaluated ILS with extended time budgets of 20s, 50s, and 200s (partially), as shown in the following table. The results demonstrate that, while ILS indeed finds slightly better solutions as time increases (10s -> 200s), the improvement remains marginal, and it still does not match the performance of our proposed method, which, in addition, exhibits significantly higher efficiency.

Table 5: Performance comparison on OPTWVP under different problem sizes and time window settings. **Bold** values denote the best performance in each setting. Category **C** refers to NCO methods, and **H** refers to heuristic baselines. The default unit for runtime is milliseconds (ms).

| Cat. | Method | $n = 50$, TW $= 100$ | | | $n = 100$, TW $= 100$ | | |
| --- | --- | --- | --- | --- | --- | --- | --- |
| | | Score ↑ | Gap ↓ | Runtime ↓ | Score ↑ | Gap ↓ | Runtime ↓ |
| – | B&C | 15.2 | 0.00% | 200 | 26.1 | 0.00% | 1023 |
| H | ILS (10 s) | 14.5 | 4.34% | 2109 | 24.9 | 4.2% | 7122 |
| | ILS (20 s) | 14.5 | 4.34% | 2120 | 24.9 | 4.2% | 7162 |
| | ILS (50 s) | 14.5 | 4.34% | 3620 | 25.0 | 4.1% | 12438 |
| | ILS (200 s) | - | - | - | - | - | - |
| C | **DeCoST (Ours)** | **15.1** | **1.06%** | 94.0 | **25.6** | **1.97%** | 158 |

| Cat. | Method | $n = 50$, TW $= 500$ | | | $n = 100$, TW $= 500$ | | |
| --- | --- | --- | --- | --- | --- | --- | --- |
| | | Score ↑ | Gap ↓ | Runtime ↓ | Score ↑ | Gap ↓ | Runtime ↓ |
| – | B&C | 51.9 | 0.00% | 1346s | – | – | >24h |
| H | ILS (10 s) | 47.8 | 7.82% | 1262 | 85.0 | – | 8441 |
| | ILS (20 s) | 47.8 | 7.82% | 1270 | 83.5 | - | 8305 |
| | ILS (50 s) | 47.8 | 7.82% | 2474 | 83.7 | - | 12407 |
| | ILS (200 s) | 47.8 | 7.82% | 8606 | 83.7 | - | 19045 |
| C | **DeCoST (Ours)** | **51.4** | **0.83%** | 60.4 | **88.1** | – | 100 |

## B.2    DISTRIBUTIONAL CHARACTERISTICS / ROBUSTNESS OF PERFORMANCE ACROSS INSTANCES

The following table reports the characteristics of the gap distribution in four different datasets (n=50,tw=100; n=50,tw=500; n=100,tw=100; n=500,tw=100).

Table 6: Distributional characteristics of gap performance

| Method | mean | std | median | q25 | q75 | min | max |
| --- | --- | --- | --- | --- | --- | --- | --- |
| POMO | 31.946 | 8.432 | 31.666 | 26.318 | 37.363 | 0.000 | 76.748 |
| Greedy-PRS | 15.904 | 7.302 | 15.343 | 10.740 | 20.395 | 0.000 | 61.360 |
| GFACS | 11.728 | 7.300 | 11.210 | 5.597 | 16.111 | 0.000 | 34.373 |
| ILS | 4.465 | 4.111 | 3.743 | 1.207 | 6.548 | 0.000 | 31.978 |
| DeCoST | **2.172** | **2.131** | **1.830** | **0.184** | **3.393** | **0.000** | **13.070** |

As shown in the table, our method DeCoST not only achieves the lowest mean value (2.172), but also demonstrates the smallest standard deviation (2.131) among all methods, indicating its overall superior performance and stability. Compared to other baselines, DeCoST yields more stable and consistent results across different instances, as reflected by its smaller interquartile range (q25–q75) and narrower minimum–maximum span.

## B.3    EXTENSION ON TEAM OPTWVP

For OPTWVP, the challenge lies in the complex coupling between discrete path selection and continuous service time allocation, which significantly expands the search space, making it difficult to quickly find high-quality solutions.

Team OPTWVP shares the similar challenge as the OPTWVP. As shown in Table 7, the additional dimension of considering multiple vehicles does not change the core challenge of the discrete-continuous coupling inherent in the problem. Therefore, our method is naturally generalisable to the team problem extension.

Table 7: Performance comparison on TOPTWVP under $n = 50$, TW $= 100$ settings. **Bold** values denote the best performance in each setting, and underlined values denote the second-best. Category **C** refers to NCO methods, and **H** refers to heuristic baselines.

| Cat. | Method | $n = 50$, TW $= 100$ Score ↑ | $\Delta$ (%) ↑ |
|---|---|---|---|
| – | B&C | - [a] | |
| H | ILS | 31.83 | -11.27 |
| C | POMO | 35.87 | - |
| C | **DeCoST (Ours)** | **36.99** | **+3.12** |

[a] Not applicable as the runtime exceeds one hour per instance.

## B.4 SENSITIVITY ANALYSIS

We conducted a comprehensive sensitivity analysis to evaluate the robustness of our method with respect to the parameters of the cost function $\beta_1$ and $\beta_2$ (default is 1000, 1000). Both parameters were systematically varied across the range [100, 500, 1000, 2000, 5000], and the resulting performance gaps are summarised in the table below. Across all combinations tested, the performance gap

Table 8: Sensitivity analysis of DeCoST with $n = 50$, TW $= 100$

| $\beta_1/\beta_2$ | 100 | 500 | 1000 | 2000 | 5000 |
|---|---|---|---|---|---|
| 100 | 0.95% | 0.88% | 0.83% | 0.88% | 0.92% |
| 500 | 0.91% | 0.96% | 0.89% | 0.83% | 0.84% |
| 1000 | 0.87% | 0.92% | 0.84% | 0.96% | 0.88% |
| 2000 | 0.92% | 0.81% | 0.84% | 0.97% | 0.84% |
| 5000 | 0.88% | 0.93% | 0.83% | 0.92% | 0.94% |

remains within the narrow range of [0.81%, 0.97%]. The mean performance gap over the 25 tested combinations is 0.89%, with a standard deviation of 0.05%, demonstrating that our method is stable and largely insensitive to moderate changes in the cost function parameters. This robustness supports the reliability of our approach under varying cost configurations.

## B.5 PERFORMANCE EVALUATION OF THE INITIAL SERVICE TIME

To evaluate the impact of initial service time settings on the quality of solution construction, experiments are conducted by varying the service time reserved in the first stage from 0% to 100% of the maximum service times in increments of 10%. In the first stage, a feasible trajectory is obtained with a fixed service time ratio. In the second stage, service times are further optimized with the Gurobi solver. Experiments are conducted on different test sets to ensure sufficient diversity in pTAR values on different trajectories. The average reward of this decomposition strategy, in comparison to the optimal reward, is illustrated in Figure 4. The results show that, on average, reserving approximately 70% of the maximum service time leads to near-optimal solutions, with an average optimality gap of 3.87% across the test sets. However, a significant number of outliers still occur in this setting. Similar issues are observed at other service time ratio levels, indicating that any fixed pre-reserved service time ratio is not robust enough across diverse instances. This justifies the necessity of designing STD and STO in solving OPTWVP.

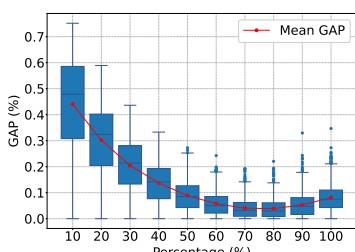

Figure 4: Analysis of initial service time ratio versus optimality gap.

### B.6 EFFICIENCY ANALYSIS OF PTAR LOSS

To investigate this, we conducted supplementary experiments on small- to medium-scale instances (n=50/100, tw=100) to see the efficiency of the introduction of pTAR loss. We define pTAR efficiency as: pTAR efficiency = the percent increase in performance gap reduction / the percent increase in runtime". This metric quantifies the "gain per unit time cost" when introducing the pTAR loss.

| Algorithm / num of nodes | gap of n50 | runtime of n50 | gap of 100 | runtime of 100 |
|---|---|---|---|---|
| DeCoST without pTAR | 2.28% | 21.48 | 3.06% | 45.71 |
| DeCoST | 1.06% | 25.67 | 1.22% | 52.55 |
| Percent increase | 53.5% | 19.5% | 60.1% | 15.0% |
| pTAR efficiency | 2.74 | | 4.0 | |

We observe that introducing the pTAR loss consistently improves performance, albeit with an additional runtime overhead of approximately 17%. Moreover, the pTAR efficiency for n=100 is higher than the pTAR efficiency for n=50, suggesting that pTAR becomes more cost-effective as instance size grows within the tested range.

## C MATHEMATICAL FORMULATION OF OPTWVP

In this paper, we mainly consider an orienteering problem (OP) variants that have a strong coupling of discrete and continuous variables, namely the orienteering problem with time windows and variable profit (OPTWVP). The mathematical formulation of OPTWVP is expressed as:

$$
\begin{aligned}
\max \quad & \sum_{i=1}^{n} p_i d_i, \\
\text{s.t.} \quad & \sum_{j=2}^{n} x_{1j} = 1 \\
& \sum_{i=1}^{n-1} x_{i|N|} = 1 \\
& y_k \le 1, \quad \forall k = 2, \dots, (|N|-1) \\
& \sum_{i=1}^{|N|-1} x_{ik} = \sum_{j=2}^{|N|} x_{kj} = y_k, \quad \forall k = 2, \dots, (|N|-1); \\
& \tau_i^- \le s_i \le \tau_i^+; \quad \forall i = 1, \dots, |N|; \\
& s_i + d_i + t_i - s_j \le L(1 - x_{ij}), \quad \forall i,j = 1, \dots, |N|; \\
& 0 \le d_i \le d_{i\,\max},
\end{aligned}
\tag{22}
$$

where $x_{ij}, y_i$ are the binary indicators whether the edge $e_{ij}$ and vertex $v_i$ is in the solution. $s_i, d_i, \tau_i^-, \tau_i^+$ denotes the start time, the service time, and time windows of the vertex $v_i$. And $\tau_i^-, \tau_i^+$, and $t_i$ denote the time window of vertex i and the travel time, respectively.

It is worth noting that OP, OPVP, OPTW can be considered as a special case of OPTWVP Marzal & Sebastia (2024). As shown in Table 9, OPTWVP includes all constraints considered in the OPTW and OPVP.

Table 9: Comparison of OP variants

| OP Variants | Time Window Constraints | Time-Varying Profits |
|---|---|---|
| OP | ✗ | ✗ |
| OPTW | ✓ | ✗ |
| OPVP | ✗ | ✓ |
| OPTWVP | ✓ | ✓ |

## D    NETWORK ARCHITECTURE OF DeCoST

The proposed DeCoST adopts an encoder-decoder architecture tailored for hybrid routing problems that involve both discrete and continuous decision variables. The model consists of a unified graph encoder and two parallel decoders: a routing decoder for node selection and a service time decoder (STD) for predicting continuous service durations.

### D.1    INPUT REPRESENTATION

Each problem instance is formulated as a fully connected graph $\mathcal{G} = (V, E)$, where the vertex feature is defined as:

$$X_i = \{x_i, y_i, s_i^-, s_i^+, p_i\}, \quad i \in V, \tag{23}$$

where $(x_i, y_i)$ denotes the spatial coordinate, $s_i^-, s_i^+$ are the start and end of the time window, and $p_i$ is the unit profit at node $i$. The edge feature is given by:

$$E = \{e_{i,j} \mid i, j \in V\}, \tag{24}$$

where $e_{i,j}$ represents the travel time or Euclidean distance between node $i$ and node $j$.

### D.2    ENCODER

The encoder transforms node and edge features into latent representations $\{h_i\}_{i \in V} \in \mathbb{R}^d$ using a stack of $L$ Transformer layers. In contrast to POMO Kwon et al. (2020), which operates solely on node embeddings, we incorporate edge features (e.g., pairwise node distances) as additive bias terms in the attention computation. This design choice is motivated by the fact that geometric relationships between nodes play a critical role in routing problems. However, conventional attention mechanisms cannot typically explicitly encode such structural information.

To address this, we adopt the spatial encoding strategy inspired by Graphormer Ying et al. (2021), and inject edge-aware bias into the attention scores. This allows the model to explicitly capture the cost and connectivity structure between nodes, thereby improving its ability to model path feasibility and graph-aware decision-making.

Formally, we modify the attention score between node $i$ and node $j$ as:

$$\alpha_{i,j} = \frac{(q_i)^T k_j + b_{i,j}}{\sqrt{d_k}}, \tag{25}$$

where $q_i$ and $k_j$ denote the query and key vectors, $d_k$ is the dimensionality of the key, and $b_{i,j}$ is an edge bias term derived from the edge feature $e_{i,j}$ (e.g., the travel distance between $i$ and $j$). We compute the edge bias for each attention head via a learnable projection:

$$b_{i,j}^{(h)} = W_{\text{edge}}^{(h)} \cdot e_{i,j}, \tag{26}$$

where $W_{\text{edge}}^{(h)} \in \mathbb{R}$ is a learnable scalar weight for the $h$-th attention head.

The final multi-head attention is then computed as:

$$\text{Attn}_h(q_i, k_j, v_j) = \sum_{j=1}^{n} \text{Softmax}\left(\alpha_{i,j}^{(h)}\right) \cdot v_j. \tag{27}$$

where $q_i, k_j, v_j$ are the linear projections of node embeddings, and $b_{i,j}$ is the attention bias derived from $e_{i,j}$.

### D.3    DECODERS

The decoder includes two parallel heads. The routing decoder computes the probability of selecting the next node as:

$$p_j = \text{Softmax}\left(\xi \cdot \tanh\left(\frac{h_a^\top h_j}{\sqrt{d}}\right)\right), \tag{28}$$

where $h_a$ is the attention output computed from the embedding of the last visited node and the current context, and $\xi$ is a temperature parameter controlling exploration. The STD shares the same structure but uses a Sigmoid activation to produce a normalized scalar $\delta_j \in [0, 1]$, representing the ratio of the maximum allowable service time allocated at node $j$. Both decoders share the same encoder output as key-value memory and are conditioned on a context vector derived from the partial solution $\tau_t$. This enables context-aware prediction of both discrete actions and continuous variables.

# E  EXPERIMENT DETAILS

## E.1  BASELINES

- Gurobi Gurobi Optimization, LLC (2024): Gurobi is a general-purpose optimization solver that implements the Branch and Cut (B&C) algorithm to obtain the exact optimal solution.

- Greedy-Profit Ratio Search (Greedy-PRS): Greedy-PRS is a two-stage customized greedy heuristic. In the first stage, the route is determined by selecting the node with the highest profit per time unit at each step. The profit per unit time is defined as the total profit obtained upon completing the node, divided by the travel time to the node plus the waiting and service time at the node. In the second stage, service times are optimized using PuLP Mitchell et al. (2011), a linear programming module that is able to solve linear programming problems, without altering the route.

- Incremental local search (ILS) Marzal & Sebastia (2024), a state-of-the-art metaheuristic search algorithm solving OPTWVP. It implements fast first solution and local search techniques by first generating an efficient initial feasible solution and then performing neighborhood search for optimization.

- POMO Kwon et al. (2020): POMO is a baseline end-to-end constructive solver that leverages the symmetry of combinatorial optimization problems to improve search efficiency. However, the original POMO framework cannot integrate edge features and handle continuous-time variables. Therefore, we extend the network to the graphformer Ying et al. (2021) and implements our proposed service time decoder (STD) and service time optimization (STO) algorithm.

- GFACS Kim et al. (2025): GFACS is a non-autoregressive generative method. It first utilizes GFlowNet to extract feature information from graph edges and nodes, and then generates heatmaps to guide the Ant Colony System for solution search. We implemented the STD and greedy/STO approaches on top of this framework.

## E.2  HYPERPARAMETERS

Table 10: Hyperparameters used for POMO and DeCoST under different dataset configurations

| Dataset Config | Train episodes | Test episodes | Epoch | Batch size | POMO size |
|---|---|---|---|---|---|
| $n = 50$, TW $= 100$ | 10000 | 10000 | 800 | 128 | 50 |
| $n = 50$, TW $= 500$ | 10000 | 100 | 800 | 128 | 50 |
| $n = 100$, TW $= 100$ | 10000 | 1000 | 800 | 128 | 50 |
| $n = 100$, TW $= 500$ | 10000 | 100 | 800 | 128 | 50 |
| $n = 500$, TW $= 100$ | 10000 | 100 | 800 | 64 | 50 |
| Solomon | 29 | 10 | 1000 | 2 | 250 |

We follow the original settings in GFACS Kim et al. (2025). For ILS Marzal & Sebastia (2024), to balance solver efficiency and ensure fair comparison, we set the maximum search time to 10 seconds during comparative experiments. For the hyperparameters listed in Table 10 are used for both the ablation studies and the comparative experiments between POMO and our proposed method, DeCoST. These settings are consistently applied across different dataset configurations to ensure a fair and reliable comparison. The additional $\beta_1, \beta_2$ hyperparameter, which balances the contribution between the primary objective loss and the $p_{\text{tar}}$ loss, is specific to DeCoST and is set to 1000, 1000, respectively. During training, the loss is small, which leads to a decrease in the gradient signal, so we magnify the loss by 1000 times to improve the training efficiency and convergence speed.

