# OpenReview forum: "Learning to Solve Orienteering Problem with Time Windows and Variable Profits"
_ICLR.cc/2026/Conference — ICLR 2026 Poster_

### Official Review · Reviewer_u59i · 2025-10-28

**Soundness:** 3
**Presentation:** 3
**Contribution:** 3
**Rating:** 6
**Confidence:** 4

**Summary:**

The paper develops a two-stage algorithm for solving the OPTWVP, where the first stage determines the order nodes are visited and the second stage determines how much time to spend at each node.  The first stage is solved heuristically using learned policy and the second stage is solved exactly using a linear programming model. Paper develops solid experimental evidence that the approach works well in practice.

**Strengths:**

1.	Solid experimental evidence that the approach is faster than state of the art methods, including global optimization and local optimization.
2.	Relatively clear exposition of the main ideas in the core paper, backed up by details in the supplemental material

**Weaknesses:**

1.	The discussion on the benchmark problems is limited, which it makes it unclear how narrow the results are.  In other words, what is different about each problem in the training/testing data set (price, travel time, etc.). If the benchmarks all share very similar structure, the results become less compelling, but are very compelling if the problem data has a lot of variance with the static space of a <TW, n> class.
2.	The practical motivation could be explicit and direct for how problems of interest map into the OPTWVP model.  It may not be readily obvious to portions of the ICLR community not versed in the orienteering/vehicle routing community

Minor comment (not a weakness) - marzal/sebastia reference appears twice in the reference list.

**Questions:**

1.	Since the second stage is a linear program, what is the advantage of using Algorithm 1 over an off-the-shelf linear programming solver, e.g. Gurobi?
2.	Can you provide more details on the OPTWVP benchmarks? Are these standard benchmarks akin to the Solomon ones mentioned later in the paper? If so, can you provide a reference for them? Or are they generated in some fashion?  If so, what parameters are varied and how?
3.	How does the model behave when presented with a completely new benchmark class, e.g. one that is not part of the training data.  In other words, if the model was only trained on n = 50 and n = 100, how do the results on n = 500 change?

---

> ### Author Response · Authors · 2025-11-20
> **Comment 1/2**
>
> Thank you very much for reviewing our paper and providing such helpful feedback. We are glad that you found our work solid and clearly presented.
> ### **W1: [limited discussion on the benchmark problems]**
> Thank you for the insightful suggestion.
> Our benchmark suite actually includes two distinct types of datasets:
> (1) randomly generated instances, and (2) Solomon-based instances.
>
> For the random datasets, the differences arise from varied values of n and TW parameters, including
> <n=50,tw=100>, <n=50,tw=500>, <n=100,tw=100>, <n=100,tw=500>, <n=500,tw=100>.
>  We have reported the detailed generation process in our response to Q2, we highlight here the most important point:
>
> It is important to note that for a fixed n, the maximum time budget is also fixed. Therefore, varying the TW size directly influences the trade-off between travel time and service-time allocation (pTAR), which creates structural differences across datasets. Smaller TW values lead to tight temporal constraints and force the solver to focus more on travel efficiency, whereas larger TW values yield a broader set of feasible service-time assignments.
>
> This creates meaningful structural variability across instances, rather than benchmarks that share a single narrow pattern.
>
> ### **W2: [Clearity]**
> Thank you for the valuable suggestion.
>
> In our revised manuscript, we have updated the first two paragraphs of the Introduction and added an explicit explanation of the Figure 1 example. These revisions more clearly articulate how real-world problems map to the OPTWVP formulation, as well as how OPTWVP relates to classical VRP variants. We believe these additions make the practical motivation more accessible to readers who may not be familiar with the orienteering or vehicle routing literature.
>
> ### **Comment: [reference appears twice]**
> We appreciate the reviewers' careful observation and have fixed the issue in the revised manuscript.
>
> ### **Q1: [what is the advantage of using Algorithm 1 over an off-the-shelf linear programming solver, e.g. Gurobi?]**
> Our method STO is explicitly designed to process a large number of STO instances in parallel, while traditional LP solvers like Gurobi are inherently instance-centric. Therefore, STO has a clear computational advantage in the training and inference process, where batch process is considered.
> As a result, STO offers a substantial computational advantage during both training and inference, which is essential for Learning-based solvers.
>
> Below we report the average runtime comparison between Gurobi and STO under different batch configurations for solving the same LP problem (with at least 15 samples per setting), which further demonstrate STO's efficiency of parallel computation:
>
> | Method \ Num of nodes        | [10,20) | [20,30) | [30,40) | [40,50) | [50,60) | [60,70) |
> |---------------|---------|---------|---------|---------|---------|---------|
> | Gurobi (b=1)  | 4.24    | 7.75    | 9.24    | 11.24   | 13.05   | 15.20   |
> | STO (b=1)     | 1.05    | 3.97    | 8.55    | 13.29   | 18.76   | 25.07   |
> | STO (b=8)     | 0.23    | 0.65    | 1.33    | 2.21    | 3.06    | NaN     |
> | STO (b=16)    | 0.11    | 0.33    | 0.67    | 1.21    | 1.76    | 2.22    |
> | STO (b=64)    | 0.05    | 0.09    | 0.20    | 0.33    | 0.49    | 0.58    |
>
> Although STO has a higher runtime than Gurobi at batch size = 1 when n > 40, its runtime decreases steadily as the batch size increases. This demonstrates that our STO algorithm method benefits substantially from batching and scales more efficiently under parallel inference.

---

> ### Author Response · Authors · 2025-11-20
> **Comment 2/2**
>
> ### **Q2: [More details on OPTWVP benchmarks]**
> Although this line of research has been studied primarily with metaheuristics [1-4], to the best of our knowledge, there are very few open-source implementations or publicly available datasets. As a result, there is no stanard benchmark dataset currently available.
>
> In this paper, the instance generation procedure is inspired by the one used in routing problems [5,6]. Given a problem of size N, we first generate a visiting
> trajectory $\tau = \{v_0, v^{(1)}, \cdots, v^{(N-2), v_0}\}$, where
> $v_0$ is the depot and intermediate nodes $v^{i}$ are randomly
> shuffled.
> The maximum service time is considered as tw/4.
> The time window is generated based on a uniform distribution: $s_i^- \sim \mathcal{U}[\psi_i - tw/2, \psi_i]$ and $\quad s_i^+= s_i^- + tw/4$, where $\psi_i$ denotes the travel time from $v_0$ to vertex $i$ of the trajectory $\tau$, and $tw$ is the width of time window.
> The maximum time of the OPTWVP instances is treated as the expected length of an arbitrary trajectory on N nodes [7], and the unit profit of the node is randomised from $p_i \sim \mathcal{U}[0, 10]$.
>
> In fact, even the Solomon-based dataset we use is generated by the ILS work itself [4]. The original Solomon dataset contains only time-window constraints, and [4] explicitly adds the variable-profit component. As stated in their paper: "Given that there is not a benchmark for this problem, a set of 56 TOPTW Solomon (set c) instances of vehicle routing problems with time windows were selected and adapted to the TOPTWVP by adding the interval of service time; in particular, $d_{min}$ was set to the service time indicated in the original problems, whereas $d_{max}$ is set to $d_{min}+30$. All these problems have a time budget of 1236 time units, and 100 nodes with different opening hours, score and service time."
>
> All our implementation details are publicly available in the hope that it will serve as a useful resource and foster further advances in both the NCO and COP communities.
>
> [1] E. Nikzad et al., Home healthcare staff dimensioning problem for temporary caregivers: A matheuristic solution approach, Comput. Oper. Res., 2023. \
> [2] D. Canca et al., Arrival and service time dependencies in the single- and multi-visit selective traveling salesman problem, Comput. Oper. Res., 2024. \
> [3] Q. Yu et al., A matheuristic approach to the orienteering problem with service time dependent profits, Eur. J. Oper. Res., 2019. \
> [4] E. Marzal et al., Solving the tourist trip design problem with time windows and variable profit using incremental local search, Appl. Soft Comput., 2024.\
> [5] Y. Kwon et al., POMO: Policy optimization with multiple optima, Neurips, 2020. \
> [6] R. Ferreira et al., A General VNS heuristic for TSPTW, Discrete Optim., 2010. \
> [7] J. Chen et al., Looking Ahead to Avoid Being Late, ArXiv, 2024.
>
> ### **Q3: [How does the model behave when presented with a completely new benchmark class]**
> To assess generalization of our approach, we applied the compared models of DeCoST and POMO trained only on the setting <n=50, tw=100>， to completely different <n,TW> settings.
>
> The results are summarized in the table below:
>
> | Test Setting   | Score (DeCoST) | Gap (DeCoST) | Score (POMO) | Gap (POMO)| Score (Origin Model) | Gap (Origin Model) |
> |-------|--------|------|-----|----|----------|---------|
> | n=50, TW=500   | 41.3| 20.07%| 32.5 | 36.7% | 51.4 | 0.83%|
> | n=100, TW=100  | 22.6| 13.12%| 18.4 | 29.5% | 25.6 | 1.97%|
> | n=100, TW=500  | 82.0| –| 51.0 | - | 88.1| –|
> | n=500, TW=100  | 38.5| 53.14%| 2.82 | 96.5% | 79.6 | 3.31%|
>
> For DeCoST, they exhibit moderate performance degradation when evaluated on the nearby settings <n=50,TW=500>, <n=100,TW=100>, and <n=100,TW=500>. The performance gap remains within 20%, indicating that the learned policy still transfers well to problems of slightly larger scale or larger time windows. However, when the model is tested on the significantly different dataset <n=500,TW=100>, the gap increases to 53.14% for DeCoST.
>
> This reveals that the learned models generalize well to nearby
> (n,TW) settings, while increasing the problem size significantly inevitably causes performance degradation since the large structural differences between datasets.

---

### Official Review · Reviewer_wvRc · 2025-10-31

**Soundness:** 3
**Presentation:** 2
**Contribution:** 2
**Rating:** 2
**Confidence:** 3

**Summary:**

This paper addresses the Orienteering Problem with Time Windows and Variable Profits (OPTWVP), a routing problem where the solution may serve any subset of nodes, subject to time-windows constraints on when each node can be served and a constraint on the duration of route, maximizing the variable profit collected from serving the nodes. The paper proposes a computational approach that consists of two stages: first, a learned policy constructs an initial solution (route and initial service times); second, a polynomial-time algorithm finds optimal service times (given the route). The policy is learned using REINFORCE with baseline, and it is based on a graph-transformer architecture. The polynomial-time algorithm for improving the solution is proven to be optimal. The proposed approach is evaluated on benchmark problem instances, demonstrating that it outperforms baselines.

**Strengths:**

* The polynomial-time algorithm for finding service times for a given route is a nice contribution, and it is proven to be optimal.
* The combination of the RL policy with the polynomial-time algorithm is beneficial (while there exist approaches that use traditional optimization methods to improve solutions provided by NCO for similar problems, there is novelty in the specific combination, which decouples the routing from the scheduling).
* The numerical results are promising.

**Weaknesses:**

* The key contribution of the paper seems to be polynomial-time algorithm for finding service time. While this is an interesting contribution, ICLR might not be the best venue for publishing it since it is not a learning-based approach. A broader AI conference might be a better fit.
* The significance of this contribution is limited by the fact that the problem can also be solved by a straightforward LP, which also takes polynomial time. So this contribution really only seems important to communities that deeply care about this optimization problem.
* The novelty in the combination of the RL policy with the polynomial-time algorithm seems to be OPTWVP-specific (generally, the idea of combining NCO with traditional optimization is not novel, but specific combination based on fixing the route is), which might be of limited interest to ICLR.
* The discussion on lines 72 to 76 seems misleading. The references argue that transformers are ill-suited for continuous time series. However, this paper considers a discrete set of variables (scheduled times) that may have continuous values, which is very different, and which should not be an issue for transformers. The paper does not make it clear what the supposed issue with "hybrid decision variables" is. In the end, the proposed RL-based approach is a fairly straightforward one (graph transformer with discrete and continuous action heads); so there does not seems to be a "hybrid decision" challenge.

The presentation of the paper needs to be revised to improve its readability. The following is a non-exhaustive list of major and minor issues:
* The paper should provide at least an informal definition of OPTWVP in the introduction for readers who are not familiar with the problem. There is currently no definition or explanation.
* References should be \citep instead of \cite when the names of the authors are not used as subject or object.
* Figure 1(a) and its description are not particularly clear, e.g., it is not obvious how the problem is mapped to OPTWVP. Perhaps present a more straightforward, vehicle-routing example.
* The features and architectures of the learned policy are not described in the main text in adequate detail. Without reading the appendix, it is not clear how the Routing Decoder and STD work together. There is limited space in the main text, but these seem like important details.
* Notation is inconsistent. On line 199, $P$ and $C$ are introduced, but they are later typeset in calligraphic font $\mathcal{P}$ and $\mathcal{C}$. On line 204, subscript $5n$ is never explained (why 5? what is $n$?).
* The constraint on line 213 does not seem to make sense. What does the expectation of inequality mean? Later, it seems that the constraints are strict, so what is the point of the expectation?
* Also, most of the notation introduced in lines 199 to 214 is never used later, so what is the point of introducing them?
* On line 224, $G$ and $M$ are never introduced or explained. Typo for $\mathcal{G}$? Batch size $M$?
* Equation (1) would be simpler if it used the notation $R(\tau_i | G)$, which is introduced anyway.
* Lines 226 to 228 discuss coupling again. The importance of this is unclear; this "coupling" is just a mixed continuous-discrete action space.

**Questions:**

* Can you elaborate on the significance of the polynomial-time algorithm for finding service time considering that an LP can also solve this?
* What other routing problems could this algorithm be applied to? Or other contributions of the paper readily applicable to other routing problems?

---

> ### Author Response · Authors · 2025-11-20
> **Comment 1/3**
>
> We sincerely appreciate the reviewer’s constructive feedback, which has helped us improve the overall quality of the paper. We have updated the manuscript to present the problem, motivation, and contributions more clearly. We have also revised the Methodology section accordingly.
>
> ### **W1: [The key contribution of the paper seems to be polynomial-time algorithm for finding service time, ICLR might not be the best venue for publishing it]**
> Thank you for the thoughtful comment.
>
> We would like to clarify that our key contribution is not proposing a new solver for the STO (LP) subproblem. Instead, our contribution lies in showing how a reinforcement-learning framework can decompose and solve a tightly coupled discrete–continuous optimization problem. Specifically, our method learns the routing structure while leveraging an LP module to optimally allocate service times, enabling effective learning on a problem class that is otherwise difficult to handle in SOTA NCO and metaheuristic approaches.
>
> Similar problem structures—where learning is used to solve or coordinate with classical optimization components—have been explored in recent ICLR works, including NCO-based methods [1–3] and learning-augmented MIP approaches [4–5]. Therefore, we believe the problem setting and methodology are well aligned with ICLR’s scope and recent trends.
>
> ### **W2: [The significance of this contribution is limited by the fact that the problem can also be solved by a straightforward LP]**
>
> Our STO module serves as an efficient batched optimizer for the RL policy rather than a new LP solver. It provides vectorizes LP solution steps so that hundreds of LPs can be solved in parallel on GPU, stable gradients/rewards to the RL policy during training.
>
> By leveraging this parallel computation, together with the cross-stage feedback mechanism, the proposed two-stage approach provides a global long-horizon estimation of service time in the early route decision stage, enabling us to search the route and service time efficiently. We believe this contribution also important to combinatorial optimization communities.
>
> ### **W3: [The novelty in the combination of the RL policy with the polynomial-time algorithm seems to be OPTWVP-specific]**
>
> We agree that the general idea of combining an RL policy with a classical optimization component has appeared in prior NCO work. However, our motivation arises from the fact that most existing NCO methods focus solely on routing, while the joint optimization of routing and service-time allocation remains under-explored and performs poorly in tightly coupled settings such as OPTWVP. This motivates us to propose DeCoST, a two-stage framework that explicitly decomposes OPTWVP and achieves substantially better performance.
>
> The coordination mechanism we propose is general to a broader class of mixed discrete–continuous COPs. The key novelty lies in providing a RL–LP coordination scheme that enables training on such tightly coupled structures—where simple decomposition leads to shortsighted policies and inefficient training processes.
> Importantly, this coordination mechanism is not specific to OPTWVP, but applies more broadly to mixed discrete–continuous COPs. For example, we demonstrate its extensibility to Team OPTWVP and Stochastic OPTWVP, and provide further discussion in our response to Q2.
>
> As noted in our response to W1, many recent ICLR papers [1–5] also investigate learning–optimization paradigm, further demonstrating that our problem setting and methodology fall well within the scope of the ICLR community.
>
> [1] Y. Kuang et al., Rethinking branching on exact combinatorial optimization solver: The first deep symbolic discovery framework, ICLR, 2024.\
> [2] A. Hottung et al., PolyNet: Learning diverse solution strategies for neural combinatorial optimization, ICLR, 2025.\
> [3] Y. Li et al., ML4TSPBench: Drawing methodological principles for TSP and beyond from streamlined design space of learning and search, ICLR, 2025.\
> [4] H. Liu et al., Apollo-MILP: An alternating prediction-correction neural solving framework for mixed-integer linear programming, ICLR, 2025.\
> [5] C. Zhang et al., Towards imitation learning to branch for MIP: A hybrid reinforcement learning based sample augmentation approach, ICLR, 2024.

---

> ### Author Response · Authors · 2025-11-20
> **Comment 2/3**
>
> ### **W4: [The discussion on lines 72 to 76 seems misleading. The paper does not make it clear what the supposed issue with "hybrid decision variables" is]**
>
> In the Introduction, we restate the key challenges of "hybrid decision variables," which are: Unlike classical VRPs that determine only the routing sequence, OPTWVP requires jointly optimizing the route and the service time spent at each node, because the reward depends directly on service times on vertices. A selected route dynamically shapes the feasible region of service time decisions through travel times and time window constraints, while the allocated service times, in turn, affect the obtained rewards and influence the overall route decisions. This bidirectional dependency prevents the two components from being optimized independently, causing an exponential expansion of the joint search space.
>
> While prior NCO works adopt a decomposition paradigm, they still rely on heuristic or trust-region–based local search when interacting with downstream optimization modules, which limits search efficiency and scales poorly under OPTWVP. In OPTWVP settings, such decomposition leads to shortsighted route predictions that cannot anticipate optimal service times, and the subsequent local adjustments are insufficient to correct the structural biases introduced in the first stage. Our approach provides a principled formulation and a coordinated learning–optimization mechanism that handles this interdependency more effectively.
>
>
> ### **[Presentation]**
> **[provide at least an informal definition of OPTWVP]**
> Thank you for the suggestion. We have updated the first and second paragraphs of Introduction to include both the motivation and a clearer description of the OPTWVP problem.
>
> **[References format]**
> Thank you for the suggestion. We have updated all corresponding references throughout the manuscript to ensure consistency and accuracy in the citation format.
>
> **[Figure 1(a) and its description are not particularly clear]**
> Thank you for the suggestion. We have added additional explanation and incorporated relevant references [6] in Figure 1 (a) to better illustrate and contextualize the problem.
>
> [6] S. Wang et al., Towards region-based robotic machining system from the perspective of intelligent manufacturing: A technology framework with case study, J. Manuf. Syst., 2023.
>
> **[The features and architectures of the learned policy are not described]**
> We thank the reviewer for this helpful suggestion. We have added additional descriptions in the main text to clarify how the route decoder and STD interact, making the architecture more understandable without requiring the reader to look at the appendix.
>
> **[Notation is inconsistent. On line 199, p and C are introduced, but they are later typeset in calligraphic font p and c. On line 204, subscript 5n is never explained (why 5? what is n?)]**
> We thank the reviewer for the careful catch. We have corrected the typos. On line 204, 5 denotes the number of constraint functions defined for each vertex, we now rewrite it as a variable $\kappa$. n denotes the number of vertices in the graph $\mathcal{G}$, we have clarified it in the formulation
>
> **[The constraint on line 213]**
> We thank the reviewer for the catch. The constraints are indeed strict, and the expectation operator was inappropriate in this context. We have removed the expectation in the revised manuscript.
>
> **[most of the notation introduced in lines 199 to 214 is never used later]**
> Thank you for the suggestion. We have removed several unnecessary modeling components while retaining the essential formulation needed to model OPTWVP as a CMDP, which is required for applying our RL-based framework. The relevant notations have been integrated into the Methodology section to better connect the problem formulation with the proposed approach.
>
> **[On line 224, G and M are never introduced or explained]**
> We thank the reviewer for pointing this out. G is the typo of $\mathcal{G}$, M is the batch size. We have clarified this in the revised manuscript.
>
> **[Equation (1) would be simpler if it used the notation R(\tau_i|G)]**
> We thank the reviewer for the suggestion. We have simplified and updated Equation (1) using the recommended notation in the revised manuscript.
>
> **[Lines 226 to 228 discuss coupling again. The importance of this is unclear]**
> "Coupling" here refers to a tightly interdependent constraint structure rather than simply having both discrete and continuous decisions. This is precisely why straightforward decomposition approaches perform poorly: the discrete policy cannot evaluate routes without solving the continuous subproblem, and the continuous subproblem depends entirely on the chosen route.
>
> To avoid misunderstanding, we have revised the text to clarify that our use of "coupling" refers specifically to this mutual dependence between routing feasibility and optimal continuous allocation, rather than merely the presence of mixed action types.

---

> ### Author Response · Authors · 2025-11-20
> **Comment 3/3**
>
> ### **Q1: [Can you elaborate on the significance of the polynomial-time algorithm for finding service time considering that an LP can also solve this?]**
>
> Our method STO is explicitly designed to address a practical computational bottleneck unique to Learning-based solvers:
>
> During training, our method must solve hundreds of such LPs repeatedly for different sampled trajectories.
> Our method STO is explicitly designed to process a large number of STO instances in parallel, while traditional LP solvers like Gurobi are inherently instance-centric.
> As a result, STO offers a substantial computational advantage during both training and inference, which is essential for Learning-based solvers.
>
> Below, we also report the average runtime comparison between Gurobi and STO under different batch configurations for solving the same LP problem (with at least 15 samples per setting), which further demonstrates STO's efficiency of parallel computation:
>
> | Method \ Num of nodes        | [10,20) | [20,30) | [30,40) | [40,50) | [50,60) | [60,70) |
> |---------------|---------|---------|---------|---------|---------|---------|
> | Gurobi (b=1)  | 4.24    | 7.75    | 9.24    | 11.24   | 13.05   | 15.20   |
> | STO (b=1)     | 1.05    | 3.97    | 8.55    | 13.29   | 18.76   | 25.07   |
> | STO (b=8)     | 0.23    | 0.65    | 1.33    | 2.21    | 3.06    | -       |
> | STO (b=16)    | 0.11    | 0.33    | 0.67    | 1.21    | 1.76    | 2.22    |
> | STO (b=64)    | 0.05    | 0.09    | 0.20    | 0.33    | 0.49    | 0.58    |
>
> Although STO has a higher runtime than Gurobi at batch size = 1 when n > 40, its runtime decreases steadily as the batch size increases. This demonstrates that our STO algorithm method benefits substantially from batching and scales more efficiently under parallel inference.
>
> ### **Q2: [What other routing problems could this algorithm be applied to? Or other contributions of the paper readily applicable to other routing problems?]**
> First, we would like to stress that OPTWVP is already a highly expressive problem class, combining both time-window (TW) and variable-profit (VP) constraints.
> Our method is applicable to many variations of some OP problems, as stated in Appendix C, such as OPTW, OP, and OPVP. To further demonstrate applicability beyond the basic setting, we have included in the supplementary material additional experiments on the multi-vehicle variant Team OPTWVP (TOPTWVP), which introduces coordination among multiple agents.
>
> In addition, we conducted further experiments on stochastic OPTWVP (SOPTWVP), a widely studied extension incorporating edge uncertainty, which is common in real-world routing problems [7,8]. We provide preliminary results below for two settings: (n=50, TW=100) and (n=50, TW=500)
>
> ### Table: Results of Stochastic OPTWVP (stochastic edge availability)
>
> | Problem type|score|gap|
> |-|-|-|
> | OPTWVP with stochastic edge availability (n=50, TW=100) | 15.0   | 1.43% |
> | OPTWVP (n=50, TW=100)| 15.1   | 1.06% |
> | OPTWVP with stochastic edge availability (n=50, TW=500)| 51.38  | 0.96% |
> | OPTWVP (n=50, TW=500)| 51.4   | 0.83% |
>
> As shown in the table, the agent’s performance does not significantly degrade under stochastic edge availability, suggesting that our method is robust to uncertainty and noise.
>
> We believe that the multi-agent coordination and stochastic uncertainty represent two of the most general and widely studied directions in meta-heuristics and NCO research, and thus provide evidence of the method’s generality in solving coupled discrete–continuous problems.
>
> That said, we acknowledge that additional work is required to fully establish transferability to other combinatorial optimization problems, and we view this as an important direction for future research.
>
> [7] L. Evers et al., A two-stage approach to the orienteering problem with stochastic weights, Comput. Oper. Res., 2014.
> [8] S. Carpin et al., Solving stochastic orienteering problems with chance constraints using Monte Carlo tree search, IEEE CASE, 2022.

---

### Official Review · Reviewer_HfEA · 2025-10-31

**Soundness:** 3
**Presentation:** 2
**Contribution:** 2
**Rating:** 4
**Confidence:** 5

**Summary:**

This paper addresses the Orienteering Problem with Time Windows and Variable Profits (OPTWVP), which involves both discrete routing and continuous service-time decisions. The authors propose a two-stage DEcoupled discrete–Continuous optimization with Service-time-guided Trajectory (DeCoST) framework. In Stage 1, a parallel decoding structure predicts a route and initial service-time allocation; in Stage 2, service times are refined via a linear programming formulation, with theoretical guarantees of global optimality. The method reportedly achieves better solution quality and faster inference than state-of-the-art constructive and metaheuristic solvers across benchmark instances.

**Strengths:**

1. Clear and structured exposition: The paper is well written, easy to follow, and methodologically consistent. The decomposition into discrete and continuous components is intuitively appealing and technically well-motivated.
2. Solid engineering contribution: The combination of a neural constructive method with an LP-based continuous-time refinement is elegant and appears to yield computational gains.

**Weaknesses:**

1. Motivation for optimizing service times unclear: In standard formulations of the orienteering problem with time windows, service times are typically derived from route structure and scheduling constraints. It remains unclear why an explicit optimization of service times is required and how this impacts practical applicability. A full formal problem statement would clarify the modeling choices.

2. Limited methodological novelty: While the decoupling approach is well-implemented, the paradigm of combining discrete RL-style route construction with an optimization-based refinement layer is well established in recent literature (e.g., in ride-hailing and hybrid combinatorial optimization works). The contribution appears primarily incremental rather than conceptually new.

3. Missing related work: The paper overlooks several closely related lines of research, in particular the Neural Search and Decision Learning frameworks developed by Kevin Tierney and co-authors. Including and contrasting these would help position the contribution more accurately.

 4. Questionable experimental evidence.
 • Table 1 raises concerns: it is surprising that Gurobi fails to find feasible solutions within 24 hours — this requires verification or clarification.
 • Figure 3 lacks subcaptions, and key plots are underexplained.
 • The absence of comparisons on standard Solomon benchmarks limits interpretability of the numerical improvements.
 • It is unclear whether the reported gains (up to 6.6×) persist across varying problem sizes and constraint tightness.

 5. Marginal theoretical contribution: Theorem 1 and Algorithm 1 formalize a well-known fact — that continuous variables over a fixed route can be optimized efficiently via LP. The inclusion adds little beyond formal completeness and could be shortened.

 6. Overstated generality: While the authors claim compatibility with various solvers and broad applicability, evidence for transferability beyond OPTWVP is not provided.

**Questions:**

1. Is it necessary to tailor algorithm 1? A standard continuous optimization problem should also be solvable in polynomial time given a fixed visit sequence?
 2. Can the authors provide stronger justification or empirical evidence for Gurobi’s failure to find feasible solutions?
 3. Why is a comaprison to Gurobi skipped for the solomon benchmark?
 4. What are the key differences between DeCoST and hybrid approaches in prior works (e.g., neural combinatorial optimization combined with LP or MILP refinement)?
 5. Would the method scale similarly for larger instances or tighter time windows?

---

> ### Author Response · Authors · 2025-11-20
> **Comment 1/3**
>
> We are grateful for the reviewer’s constructive comments, which greatly contributed to improving the clarity and quality of the manuscript. We have updated the manuscript to present the problem, motivation, related studies and contributions more clearly.
>
> ### **W1: [Motivation for optimizing service times unclear]**
> Thank you for the helpful suggestion. In the revised manuscript, we expand the first two paragraphs of the Introduction to include a more complete problem statement and clearer motivation. We also provide a detailed formulation in Appendix C for improved clarity.
>
> ### **W2: [Limited methodological novelty]**
> We acknowledge that previous works have explored the general paradigm of combining discrete RL-style route construction with an optimization-based refinement layer. However, our work differs from these approaches in two important aspects:
>
> (1) Novelty in the problem being addressed.
> OPTWVP has been widely studied in the heuristic optimization literature. However, their performances are often heavily limited by manually designed heuristics and exhaustive refinement procedures. This motivates turning to the NCO paradigm, which has already shown strong potential on VRPs. However, OPTWVP-like problem remains largely unexplored in the NCO community: Existing NCO methods focus solely on routing, while the joint optimization of routing and service-time allocation remains under-explored. The joint optimization greatly increases the search space and standard NCO architectures struggle to handle such mixed discrete–continuous structures effectively.
>
> (2) Novelty in the methodology for handling this coupling.
> While prior NCO works adopt a decomposition paradigm, they still rely on heuristic or trust-region–based local search when interacting with downstream optimization modules, which limits search efficiency and scales poorly under OPTWVP. In OPTWVP settings, such decomposition leads to shortsighted route predictions that cannot anticipate optimal service times, and the subsequent local adjustments are insufficient to correct the structural biases introduced in the first stage. Our approach provides a principled formulation and a coordinated learning–optimization mechanism that handles this interdependency more effectively.
>
> ### **W3: [Missing related work]**
> Thank you for the reminder. We have added more related work [1-9] and incorporated them into both the Introduction and Related Work sections of the revised manuscript.
>
> [1] Y. Kuang et al., Rethinking branching on exact combinatorial optimization solver: The first deep symbolic discovery framework, ICLR, 2024.
>
> [2] A. Hottung et al., PolyNet: Learning diverse solution strategies for neural combinatorial optimization, ICLR, 2025.
>
> [3] F. Berto et al., RouteFinder: Towards foundation models for vehicle routing problems, Trans. Mach. Learn. Res., 2025.
>
> [4] A. Hottung et al., Neural deconstruction search for vehicle routing problems, Trans. Mach. Learn. Res., 2025.
>
> [5] Y. Li et al., ML4TSPBench: Drawing methodological principles for TSP and beyond from streamlined design space of learning and search, ICLR, 2025.
>
> [6] H. Liu et al., Apollo-MILP: An alternating prediction-correction neural solving framework for mixed-integer linear programming, ICLR, 2025.
>
> [7] M. Paulus et al., Learning to dive in branch and bound, NeurIPS, 2023.
>
> [8] X. Hu et al., Multi-stage Predict+Optimize for (mixed integer) linear programs, NeurIPS, 2024.
>
> [9] Y. Li et al., Fast and interpretable mixed-integer linear program solving by learning model reduction, AAAI, 2025.
>
> ### **W4: [Questionable experimental evidence]**
> **[it is surprising that Gurobi fails to find feasible solutions within 24 hours in Table 1]**
>
> Gurobi is indeed highly efficient for routing-only MIP formulations. However, OPTWVP introduces an additional layer of complexity: apart from discrete routing decisions, the solver must also determine continuous service-time variables. Once service time becomes part of the decision space, the solver must effectively explore a search space on the order of $TW\_size^n$, which grows exponentially with both the number of nodes and the size of the time windows. This dramatically increases the complexity compared with classical routing MIPs.
>
> This behavior is also reflected in Tables 1 and 2 in the manuscript. When TW = 100, Gurobi can still find optimal solutions efficiently even as the problem size increases from 50 to 500 nodes. However, when TW = 500, the enlarged search space makes the problem significantly harder: the solver struggles to prune infeasible branches early, leading to extremely long search times and, in several cases, failure to find feasible solutions within 24 hours.
>
> **[Figure 3 lacks subcaptions, and key plots are underexplained]**
>
> We thank the reviewer for the suggestion. We have revised Figure 3 accordingly and added additional descriptions to Figure 1 to better introduce the OPTWVP problem.

---

> ### Author Response · Authors · 2025-11-20
> **Comment 2/3**
>
> **[The absence of comparisons on standard Solomon benchmarks limits interpretability of the numerical improvements]**
>
> We thank the reviewer for this insightful comment.
>
> The settings of the OPTWVP solved in our paper and in [10] are not fully aligned with the standard Solomon benchmark. The original Solomon dataset contains only time-window constraints and don't have the variable-profit component.
>
> In fact, Solomon-based dataset we use is generated by the ILS work itself [10]. As stated in their paper: "Given that there is not a benchmark for this problem, a set of 56 TOPTW Solomon (set c) instances of vehicle routing problems with time windows were selected and adapted to the TOPTWVP by adding the interval of service time; in particular, $d_{min}$ was set to the service time indicated in the original problems, whereas $d_{max}$ is set to $d_{min}+30$. All these problems have a time budget of 1236 time units, and 100 nodes with different opening hours, score and service time."
>
> Meanwhile, all our implementation details are publicly available in the hope that it will serve as a useful resource and foster further advances in both the NCO and COP communities.
>
> [10] E. Marzal et al., Solving the tourist trip design problem with time windows and variable profit using incremental local search, Appl. Soft Comput., 2024.
>
> **[It is unclear whether the reported gains (up to 6.6×) persist across varying problem sizes and constraint tightness]**
> We thank the reviewer for this helpful suggestion.
>
> The reported 6.6× improvement refers to runtime speedup. In Tables 1 and 2 of the manuscript, we compare runtime only against methods achieving a solution gap below 10%. As shown in Table 1, our method achieves even larger gains—20x to 100xspeedup—on smaller instances (n=50, 100), while obtaining 6.6x speedup at <n=500, TW=100>.
> We have updated the abstract to reflect this more clearly.
>
> ### **W5: [Marginal theoretical contribution]**
> Thank you for the suggestion. We have streamlined the Service Time Optimization section accordingly.
>
> ### **W6: [Overstated generality]**
> First, we would like to stress that OPTWVP is already a highly expressive problem class, combining both time-window (TW) and variable-profit (VP) constraints. To further demonstrate applicability beyond the basic setting, we have included in the supplementary material additional experiments on the multi-vehicle variant Team OPTWVP (TOPTWVP), which introduces coordination among multiple agents.
>
> In addition, we conducted further experiments on stochastic OPTWVP (SOPTWVP), a widely studied extension incorporating edge uncertainty, which is common in real-world routing problems [11,12,13,14]. We provide preliminary results below for two settings: (n=50, TW=100) and (n=50, TW=500)
>
> | Problem type|score|gap|
> |-|-|-|
> | OPTWVP with stochastic edge availability (n=50, TW=100) | 15.0   | 1.43% |
> | OPTWVP (n=50, TW=100)| 15.1   | 1.06% |
> | OPTWVP with stochastic edge availability (n=50, TW=500)| 51.38  | 0.96% |
> | OPTWVP (n=50, TW=500)| 51.4   | 0.83% |
>
> As shown in the table, the agent’s performance does not significantly degrade under stochastic edge availability, suggesting that our method is robust to uncertainty and noise.
>
> We believe that the multi-agent coordination and stochastic uncertainty represent two of the most general and widely studied directions in meta-heuristics and NCO research, and thus provide evidence of the method’s generality in solving coupled discrete–continuous problems.
>
> That said, we acknowledge that additional work is required to fully establish transferability to other combinatorial optimization problems, and we view this as an important direction for future research.
>
> [11] L. Evers et al., A two-stage approach to the orienteering problem with stochastic weights, Comput. Oper. Res., 2014.
> [12] S. Carpin et al., Solving stochastic orienteering problems with chance constraints using Monte Carlo tree search, IEEE CASE, 2022.

---

> ### Author Response · Authors · 2025-11-20
> **Comment 3/3**
>
> ### **Q1: [Is it necessary to tailor algorithm 1?]**
> We fully agree that, given a fixed visit sequence, the service-time allocation problem reduces to a standard LP, which can indeed be solved in polynomial time by conventional LP solvers.
>
> However, Algorithm 1 is introduced to address a practical computational bottleneck unique to learning-based solvers:
>
> During training, our method must solve hundreds of such LPs repeatedly for different sampled trajectories.
> Our method STO is explicitly designed to process a large number of STO instances in parallel, while traditional LP solvers like Gurobi are inherently instance-centric. Therefore, STO has a clear computational advantage in training and inference process, where batch process is considered.
>
> Below we report the average runtime comparison between Gurobi and STO under different batch configurations for solving the same LP problem (with at least 15 samples per setting), which further demonstrates STO's efficiency of parallel computation:
>
> | Method \ Num of nodes | [10,20) | [20,30)| [30,40) | [40,50) | [50,60) | [60,70) |
> |-|-|-|-|-|-|-|
> |Gurobi (b=1)|4.24|7.75|9.24|11.24|13.05|15.20|
> |STO (b=1)|1.05|3.97|8.55|13.29|18.76|25.07|
> |STO (b=8)|0.23|0.65|1.33|2.21|3.06|-|
> |STO (b=16)|0.11|0.33|0.67|1.21|1.76|2.22|
> |STO (b=64)|0.05|0.09|0.20|0.33|0.49|0.58|
>
> Although STO has a higher runtime than Gurobi at batch size = 1 when n > 40, its runtime decreases steadily as the batch size increases. This demonstrates that our STO algorithm method benefits substantially from batching and scales more efficiently under parallel inference.
>
> ### **Q2: [Stronger justification or empirical evidence for Gurobi’s failure to find feasible solutions]**
>
> Analytically, this is because OPTWVP expand the search space due to discrete routing decisions and continuous service-time allocations are tightly coupled. We update the description in the introduction of the revised manuscript:
>
> "Compared with classical VRPs, OPTWVP represents a more realistic and challenging class of VRP in which discrete routing decisions and continuous service-time allocations are tightly coupled: Unlike classical VRPs that determine only the routing sequence, OPTWVP requires jointly optimizing the route and the service time spent at each node, because the reward depends directly on service times on vertices. A selected route dynamically shapes the feasible region of service time decisions through travel times and time window constraints, while the allocated service times, in turn, affect the obtained rewards and influence the overall route decisions. This bidirectional dependency prevents the two components from being optimized independently, causing an exponential expansion of the joint search space."
>
> Empirically, our experiments also indicate that the time-window size has a larger impact on Gurobi’s runtime than the number of nodes.
>
> For convenience, you may also verify this behavior using our open-source code. A lightweight test can be conducted with the following commands:
> ```
> python generate_data.py --problem "OPTWVP" --problem_size 50 --num_samples 1 --hardness "hard" --timewindows 100   # or 500
> python compare_gurobi.py
> ```
>
> ### **Q3: [Why is a comparison to Gurobi skipped for the solomon benchmark?]**
> Thank you for the suggestion. We add the Gurobi runtime experiments on the Solomon benchmark. Unfortunately, on the same PC environment, due to the OPTWVP challenge described in response to Q2, Gurobi required more than 12 hours to solve these instances, making a full comparison infeasible within reasonable time limits.
>
>
> ### **Q4: [What are the key differences]**
> This has been discussed in reply to W2. In short,
> The prior works still rely on heuristic or trust-region–based local search when interacting with LP/MILP refinement modules, which limits search efficiency and scales poorly under OPTWVP. In OPTWVP settings, such decomposition leads to shortsighted route predictions that cannot anticipate optimal service times, and the subsequent local adjustments are insufficient to correct the structural biases introduced in the first stage.
> Our approach DeCoST decomposes OPTWVP into a routing problem and a service time allocation problem, which effectively decouples the searching process while enabling efficient and learnable coordination.
>
> ### **Q5: [scalability]**
> We have additionally included experiments of DeCoST under tighter time windows (n=50, TW=50). However, the n=1000 setting is skipped because it is highly computationally demanding, and producing stable results would require substantially more time than is currently feasible. Our method achieves strong results when the problem size is under n<500. For larger-scale instances, we believe that applying appropriate scaling techniques will be necessary, which is an interesting future direction.
> |Method|Score|Gap|Runtime (ms)|
> |-|-|-|-|
> |B&C|7.98|-|-|
> |ILS|7.66|4.01%|1633|
> |GFACS|7.85|1.65%|9223|
> |POMO|6.12|23.21%|29.1|
> |DeCoST|7.89|1.08%|66.8|

---

> > ### Comment · Reviewer_HfEA · 2025-11-27
> > **feedback**
> >
> > I thank the authors for answering in detail to all of my concerns and making changes to the manuscript accordingly. I raised my score to reflect the authors effort, which increases the quality of the work

---

> > > ### Author Response · Authors · 2025-11-28
> > >
> > > We sincerely thank the reviewer for the positive feedback and the updated assessment of our work! We are glad that our clarifications and revisions addressed your concerns. Thank you again for your constructive comments, which greatly helped improve the quality of the paper.

---

### Official Review · Reviewer_MPHD · 2025-11-08

**Soundness:** 3
**Presentation:** 3
**Contribution:** 2
**Rating:** 6
**Confidence:** 2

**Summary:**

This paper introduces DeCoST, a two-stage framework designed to solve hybrid combinatorial optimization problems that combine discrete routing and continuous time-dependent decisions. Specifically,  the discrete components refer to the routing decisions (i.e choosing which locations to visit and in what order) while the continuous components concern the time variables, such as how long to spend at each location and when to start service within the allowed time windows. The DeCoST algorithm first determines the discrete route and then optimizes the continuous service times through a linear program to maximize the overall reward. In the first stage, DeCoST employs a parallel decoder to generate a feasible route and an initial estimate of service times, enhanced by spatial encoding and feasibility masks that respect time-window constraints. A profit-weighted time allocation ratio (pTAR) is introduced in order to learn the trade-off between travel time and service time allocation for the initial service time assignment. In the second stage, given the fixed route, a linear programming (LP)-based Service Time Optimization (STO) algorithm computes globally optimal service times.
Experiments show that DeCoST consistently outperforms both heuristic and neural combinatorial optimization (NCO) baselines. Overall, DeCoST offers a theoretically grounded, efficient, and generalizable approach to hybrid discrete-continuous optimization, bridging exact optimization and neural learning methods.

**Strengths:**

The authors propose a novel well-designed two-stage framework that cleanly separates discrete and continuous decision-making for the orienteering problem with time windows and variable profits. It also combines rigorous theoretical guarantees via the LP formulation in the second stage with extensive experiments supporting the results.

**Weaknesses:**

One weakness of the paper is that, unlike the second stage, the first-stage routing policy has no theoretical guarantee of optimality. It relies on reinforcement learning, which may converge to only locally optimal routes, so the overall solution quality depends on the effectiveness of this learned policy without any formal performance bound.

**Questions:**

Is it possible to provide any form of theoretical guarantee or performance bound for the first-stage routing policy?

---

> ### Author Response · Authors · 2025-11-21
> **Comment**
>
> We sincerely thank the reviewer for the positive evaluation and for considering our work novel and well-designed.
> Below we clarify the theoretical properties of our first-stage framework.
>
> **[Local Optimality of NP-hard VRPs]**
> Since the classical VRP family is NP-hard, obtaining globally optimal solutions in polynomial time is generally infeasible. Most existing approaches therefore rely on heuristic algorithms such as ACO [2], GA [3], ILS [4], MCTS [5], or NCO [6], all of which can only guarantee locally optimal results [7].
>
> **Therefore, local minimum is an inherent property of NP-hard routing problems faced by all heuristic and learning-based solvers, rather than a limitation of our approach. **Furthermore, here we provide explicit theoretical analysis supporting its local optimality.
>
> **[Local Optimality of First Stage]**
> From an analytical perspective, our method ensures at least sub-optimality by enforcing strict feasibility through hard masking.
> Specifically, we encode all OPTWVP routing constraints into the RL model's next-node decision process, which guarantees that:
>
> 1. the trajectory is terminated when there is no enough budget for a new step
> 2. the following nodes can not be choosen if:
>       - There is no enough budget to go to the node and go back depot: $t_{cur} + t_{to next node} + t_{to depot} > \tau_{\max}$;
>       - The arrive time at the node exceed its time windows: $t_{next arrival} > TW_{end}$;
>       - The node has been visited.
>
> This ensures that all model actions satisfy OPTWVP constraints, which is also supported by our experiments.
>
> Due to the strict feasibility masking, the routing policy always produces a valid solution within the feasible region
>
> $\pi_\theta \in {\Pi_c}$
>
> Given this feasible region, the learning dynamics of policy-gradient RL ensure that the policy monotonically improves the objective inside ${\Pi_c}$：
>
> $J_r(\pi_{\theta_{t+1}}) \leq J_r(\pi_{\theta_{t}})$
>
> Therefore, upon convergence, the policy satisfies:
>
> $\pi_\theta \in {\Pi_c}$,\
> $\nabla J_r(\pi_{\theta}) =0$,
>
> which demonstrates that the policy converges to a locally optimal feasible solution.
>
> **[Extension to more general cases]**
> For more general cases, existing theory also provides local optimality guarantees for policy-gradient reinforcement learning under mild assumptions [1]. Below we outline how such guarantees apply to our routing formulation.
>
> We can extend and formulate the routing problem as a chance-constrained MDP:
>
> $\min_{\pi_{\theta}}J_r(\pi_{\theta})$
>
> $\quad \text{s.t.}\
> \Pr(J_c(\pi_{\theta})>\alpha)\le\beta$
>
> where $J_r$ is the expected cumulative cost and $J_c$ represents the cumulative constraint-related cost. $\Pr( J_c(\pi_{\theta})>\alpha)\le\beta$ limits the probability of violating the safety/risk threshold $\alpha$.
>
> Following the standard CMDP formulation, this constrained problem can be relaxed into a Lagrangian objective:
>
> $\min_{\theta}\ \max_{\lambda\ge0}\;
> L(\theta,\lambda)=J_r(\pi_\theta)+\lambda(\Pr(J_c>\alpha)-\beta),$
> which can be optimized through policy-gradient updates on $\theta$ and dual updates on $\lambda$.
>
> As shown in Section 5.1 of [1], the update rule for the primal-dual variables satisfies the required regularity conditions (differentiability, strict feasibility). Consequently, Theorem 13 in [1] establishes the convergence of this procedure to a stationary point, guaranteeing that the learned routing policy is a locally optimal feasible solution of the CMDP.
>
> Existing studies have also adopted similar Lagrangian relaxation techniques [6], further demonstrating the potential of this direction. We consider this line of work highly meaningful and plan to investigate it systematically in future work, with the aim of enhancing the overall performance and generality of our two-stage framework and providing stronger theoretical guarantees for the overall policy.
>
> We hope our responses can address the concern, and we welcome further discussions if any point is unclear!
>
> [1] Y. Chow et al., Risk-constrained reinforcement learning with percentile risk criteria, J. Mach. Learn. Res., 2018.
> [2] Y. Sun et al., Boosting ant colony optimization via solution prediction and machine learning, Comput. Oper. Res., 2022.\
> [3] F. Carrabs, A biased random-key genetic algorithm for the set orienteering problem, Eur. J. Oper. Res., 2021.\
> [4] E. Marzal et al., Solving the tourist trip design problem with time windows and variable profit using incremental local search, Appl. Soft Comput., 2024.\
> [5] S. Carpin et al., Solving stochastic orienteering problems with chance constraints using Monte Carlo tree search, IEEE CASE, 2022.\
> [6] J. Bi et al., Learning to handle complex constraints for vehicle routing problems, NeurIPS, 2024.\
> [7] C. Blum et al., Metaheuristics in combinatorial optimization: Overview and conceptual comparison, ACM Comput. Surv., 2003.

---

### Author Response · Authors · 2025-12-02
**Summary**

Dear AC and reviewers:

Thank you for overseeing the review process and for the constructive feedback on our submission.

Due to the shortened discussion period, we received only one follow-up message from Reviewer HfEA. **Reviewer HfEA was positive**, stating: "*I thank the authors for answering in detail to all of my concerns and making changes to the manuscript accordingly. I raised my score to reflect the authors' effort, which increases the quality of the work.*" We are pleased that the reviewer acknowledged that our revisions successfully addressed the previously raised issues.

During the discussion period, we responded thoroughly to all weaknesses and questions. Below is a summary of the main concerns raised by the reviewers. The reviewers' concerns are mainly in:

1. Theoretical guarantee of learning-based routing policy *(Raised by Reviewer MPHD)*
2. Unclear motivation, contribution, and related work *(Raised by Reviewer HfEA (**positive response**), wvRC, and u59i)*
3. Benchmark variations *(Raised by Reviewer u59i)*
4. Importance of Algorithm 1 for LP service time optimization *(Raised by Reviewer HfEA (**positive response**), wvRC, and u59i)*
5. Experiment details (Benchmarks) *(Raised by Reviewer HfEA (**positive response**), u59i)*
6. Generality of the proposed approach *(Raised by Reviewer HfEA (**positive response**) and wvRC)*
7. Readability of the manuscript *(Raised by Reviewer HfEA (**positive response**) and wvRC)*

We carefully replied to all of these points in our responses and included most of the discussion directly into the revised manuscript to further improve the clarity, contribution, and soundness of our work.
We hope that our replies and revisions satisfactorily address the remaining concerns and further clarify the contribution of our work.

Thank you again for your consideration.

Best regards,\
The DeCoST Authors

---

### Meta-Review · Area_Chair_3fHf · 2026-01-07

**Summary:**

- This paper proposes a novel framework for the Orienteering Problem with Time Windows and Variable Profits problem.

- Four reviews were collected, with scores of 6 6 4 2. The reviewer who gave a 4 has decided to raise the score during the discussion.

- All the reviewers appreciate the solid methodology and theoretical contributions and the promising numerical results.

- The most negative reviewer (wvRc with score 2) thinks the paper contains an interesting contribution, but doesn't think ICLR is the best venue. Also, the reviewer wvRc questions the significance of the contribution due to its reliance on a linear programming solver. These concerns were addressed successfully by the rebuttal.

**Reviewer Concerns:**

**Solved concerns**
-  Reviewer wvRc with score 2 doesn't think ICLR is the best venue. But the rebuttal clarified that similar problem structures were explored in recent iclr publications.

- Reviewer wvRc questions the the significance of the contribution due to its reliance on a linear programming solver. The rebuttal clarifies that the STO module serves as an efficient RL solver and can be solved in parallel on GPU.

- Reviewer wvRc's clarification questions are also addressed in the rebuttal.

- Reviewer  HfEA also questions the missing related work and the failure of Gurobi in the experiments. The rebuttal provides satisfactory responses to them.

- Reviewer HfEA's concerns on generality was resolved in the rebuttal by additional experiments.

- Reviewer HfEA acknowledged in the discussion that the rebuttal has addressed all their concerns.

**Reviewer Scores:**

Four reviews were collected, with scores of 6 6 4 2. The reviewer who gave a 4 has decided to raise the score during the discussion. The negative reviewer with a score 2 is likely to raise their score too.

---

### Decision · Program_Chairs · 2026-01-26

Accept (Poster)